# Mouse enteric neurons control intestinal plasmacytoid dendritic cell function via serotonin-HTR7 signaling

Hailong Zhang[1,2,3], Yuko Hasegawa[1,2,3], Masataka Suzuki [1,2,3], Ting Zhang[1,2,3], Deborah R. Leitner[1,2,3], Ruaidhrí P. Jackson[4] & Matthew K. Waldor [1,2,3] ✉

Serotonergic neurons in the central nervous system control behavior and mood, but knowledge of the roles of serotonergic circuits in the regulation of immune homeostasis is limited. Here, we employ mouse genetics to investigate the functions of enteric serotonergic neurons in the control of immune responses and find that these circuits regulate IgA induction and boost host defense against oral, but not systemic *Salmonella* Typhimurium infection. Enteric serotonergic neurons promote gut-homing, retention and activation of intestinal plasmacytoid dendritic cells (pDC). Mechanistically, this neuro-immune crosstalk is achieved through a serotonin-5-HT receptor 7 (HTR7) signaling axis that ultimately facilitates the pDC-mediated differentiation of IgA⁺ B cells from IgD⁺ precursors in the gut. Single-cell RNA-seq data further reveal novel patterns of bidirectional communication between specific subsets of enteric neurons and lamina propria DC. Our findings thus reveal a close interplay between enteric serotonergic neurons and gut immune homeostasis that enhances mucosal defense.

Serotonin, also known as 5-hydroxytryptamine (5-HT), can be produced by several cell types and has diverse functions[1]. In central nervous system (CNS) neurons, where 5-HT synthesis depends on the enzyme tryptophan hydroxylase 2 (*Tph2*)[2], serotonin acts as a neurotransmitter whose activity is associated with regulation of emotion and mood[3]. Mutations in *Tph2* have been associated with depression, anxiety, impulsivity, and aggression in both humans and mice[4,5]. Outside the nervous system, cells use *Tph1* to produce 5-HT. Studies using knockout mice have shown that this enzyme creates the majority of the serotonin found in blood and in the gut[6]. This non-neuronal derived serotonin, which is primarily produced by intestinal enterochromaffin cells, plays important roles in diverse processes, including hemostasis, vascular tone, and immune modulation[7].

While the importance of serotonin produced by neurons in the CNS in the control of emotion and mood is well-established[8], the functions of serotonin produced by enteric neurons are not as well understood. In the gut, the existence of enteric serotonergic neurons was initially proposed by Gershon et al. in 1965[9] and recent single-cell RNA sequencing data confirm this idea[10]. Enteric serotonergic neurons are thought to promote mucosal homeostasis and influence neurogenesis and gut motility[11]. However, studies of the functions of serotonin produced by enteric neurons have been hindered by the absence of conditional knock-out (KO) mice specifically deficient in *Tph2* in enteric neurons, with intact CNS *Tph2*.

Connections between neuronal serotonin and the immune system have not been experimentally investigated, despite the recent marked expansion in knowledge of the compounds and mechanisms that mediate the communication between the nervous system and immune system. Neuronal circuits are now known to influence both innate and adaptive immune responses[12]. For example, neurons within the sympathetic and vagus nerves and dorsal root ganglion as well as nociceptive neurons have been shown to regulate innate immune cells,

¹Division of Infectious Diseases, Brigham and Women's Hospital, Boston, MA 02115, USA. ²Department of Microbiology, Harvard Medical School, Boston, MA 02115, USA. ³Howard Hughes Medical Institute, Boston, MA 02115, USA. ⁴Department of Immunology, Harvard Medical School, Boston, MA 02115, USA. ✉e-mail: mwaldor@research.bwh.harvard.edu

including macrophages, mast cell, and innate lymphoid cells (ILCs)[13]. In these pathways, compounds traditionally associated with communication between neurons, such as norepinephrine (NE), calcitonin gene-related peptide (CGRP), substance P, and neuromedin U (NMU) have been shown to modify immune cell function and play roles in immune development and homeostasis and host defense[14]. Since serotonergic neuronal circuits have been strongly associated with the control of emotion/mood, we wondered whether neuronal serotonin could also modify immune responses and host defense, given the extensive reports of associations between emotion/mood and susceptibility to disease[15]. Interestingly, a recent report suggested an association between a human *Tph2* variant and sensitivity to infection[16].

Here, using mice lacking neuronal serotonin synthesis, we address whether serotonergic circuits regulate immune homeostasis and host defense. Employing whole-body *Tph2* KO and a conditional KO mouse model (*Tph2^fl/fl; Hand2-Cre*) lacking *Tph2* in intestinal neurons, we show that enteric serotonergic neurons support the gut homing and/or retention and activation of plasmacytoid dendritic cells (pDC) via serotonin - HTR7 signaling, and that defects in pDC likely underlie the reduction in IgA B cell differentiation in the small intestine in the absence of neuronal serotonin. Collectively, our findings suggest that enteric neuronal 5-HT signaling to HTR7+ pDC regulates intestinal immune homeostasis and mucosal defense.

## Results

### Serotonergic neurons regulate gut immune homeostasis and mucosal defense

To begin to address whether serotonin impacts host defense, we used mice deficient in serotonin transporters (*Sert*). In *Sert^-/-* animals, serotonin signaling is heightened due to the absence of serotonin

reuptake[17]. Following oral infection with *Salmonella enterica* serovar Typhimurium, a model enteric pathogen, *Sert^-/-* mice had decreased pathogen burdens in the spleen and liver (Supplementary Fig. 1A) and increased survival compared with co-housed littermate controls (Supplementary Fig. 1B). These findings are consistent with a role for serotonin in host defense as suggested by a previous study[18]. However, given the expression of *Sert* in several cell types besides neurons[19], these data do not provide conclusive evidence for the action of neuronal serotonin in gut defense.

To more directly address the role of neuronal serotonin in host defense, we challenged *Tph2^-/-* animals, which lack serotonin production in neurons, with *Salmonella*. These animals have often been used to model a set of emotional disorders[20,21], but the integrity of their mucosal defense has not been explored. Following oral infection with *Salmonella, Tph2^-/-* mice had ~100-fold higher pathogen burdens in the spleen and liver (Fig. 1A) and accelerated death and reduced survival compared with co-housed littermate controls (Fig. 1B). In contrast, when the pathogen was administered intravenously to WT and *Tph2^-/-* mice, there was no difference in survival and both groups succumbed to infection with similar kinetics (Supplementary Fig. 1C). Together, these observations suggest that *Salmonella* dissemination from the intestine is elevated in *Tph2^-/-* mice.

To address the possibility that serotonergic circuits modify intestinal immune cell function, we analyzed the gene expression profiles of lamina propria-derived immune cells (CD45+) in co-housed littermate WT and *Tph2^-/-* mice. In CD45+ cells derived from the small intestines of *Tph2^-/-* mice, there was reduced expression of genes linked to B cell activation compared to WT animals (Fig. 1C, D). However, the transcription profiles of CD45+ cells isolated from the colons of *Tph2^-/-* and WT mice were similar (Supplementary Fig. 1D). In contrast to the changes in gene expression profiles in small intestinal

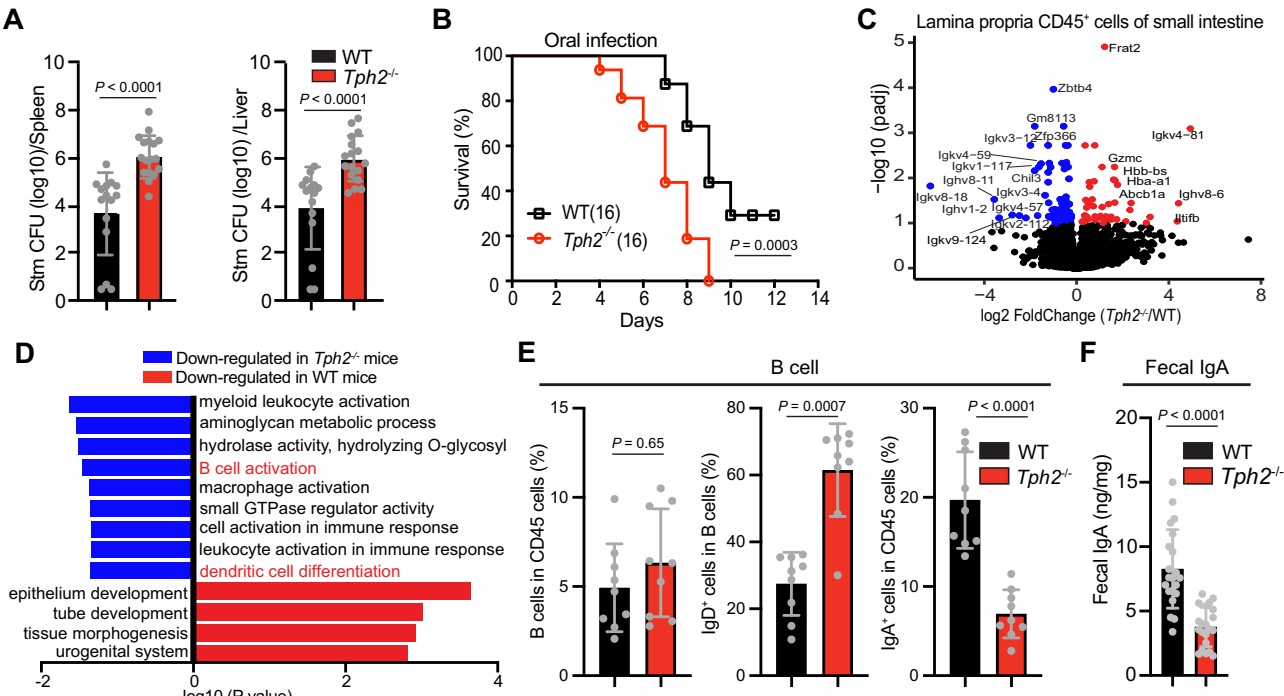

**Fig. 1 | Neuronal serotonin is important for IgA B cell differentiation and host defense against oral *Salmonella* infection. A** *Salmonella* Typhimurium (STm) CFU burdens in spleens and livers of WT and *Tph2^-/-* mice 5 days post oral inoculation (WT, *n* = 15; *Tph2^-/-*, *n* = 17). **B** Kaplan−Meier plot of survival of WT and *Tph2^-/-* mice following oral STm inoculation (*n* = 16). **C** Differential gene expression of CD45+ cells isolated from the lamina propria of the small intestine from co-housed WT and *Tph2^-/-* mice. Red points indicate increased expression and blue points indicate

decreased expression in *Tph2^-/-* mice compared to WT mice. **D** Gene Ontology pathway analysis of data from Fig. 1C. **E** Flow cytometry analyses of frequency of total B cells, IgD+ fraction of total B cells, and IgA+ cells in the lamina propria CD45+ cells from the small intestine of WT and *Tph2^-/-* mice (*n* = 9). **F** Fecal IgA levels in WT and *Tph2^-/-* mice (*n* = 21). Data shown are means ± SD. Statistical analysis was performed by two-tailed Mann−Whitney test in **A**, **E**, **F**; and by a Log-rank test in **B**.

CD45$^+$ cells that were observed in *Tph2$^{-/-}$* animals, transcriptional profiles of epithelial cells isolated from the small and large intestines of WT and *Tph2$^{-/-}$* mice were similar (Supplementary Fig. 1E, F). Thus, *Tph2*-dependent processes appear to specifically modulate gene expression of small intestinal lamina propria immune cells.

FACS-based immune profiling of CD45$^+$ cells isolated from the lamina propria of the small intestines from *Tph2$^{-/-}$* mice also revealed that serotonergic neurons modify immune cell composition in the small intestine. Even though the total fraction of B cells was similar in *Tph2$^{-/-}$* and control animals, there were increased IgD$^+$ B cells and decreased IgA$^+$ B cells in *Tph2$^{-/-}$* animals (Fig. 1E, Supplementary Figs. 1G and 2), consistent with a defect in the differentiation of IgD$^+$ to IgA$^+$ B cells. Furthermore, there was a reduction of fecal IgA in the *Tph2$^{-/-}$* mice (Fig. 1F). These data suggest that serotonergic circuits promote the development of IgA$^+$ B cells from IgD$^+$ precursors in the lamina propria of the small intestine; however, defects in B cell trafficking and recirculation cannot be excluded. The fecal microbiota in *Tph2$^{-/-}$* and control animals were similar, likely due to coprophagic behavior in the co-housed littermate animals (Supplementary Fig. 3A, B).

## Enteric serotonergic neurons control intestinal IgA B cell development

The *Tph2$^{-/-}$* animals lack serotonin production by both central and peripheral neurons. The potential immune functions of serotonin produced by intestinal neurons have received relatively little attention. Since enteric serotonergic neurons (ESN) could underlie the association between serotonergic neurons and the intestinal immune system, we initially quantified this neuronal population. FACS was used to sort cells bearing the neuronal marker *Uchl1*[22] from the small intestines of infant mice, and then single-nucleus RNA-seq was carried out to determine the frequency of *Tph2*-expressing neurons. These analyses revealed that a small proportion (~2%) of cells expressed *Tph2* as well as *Ret* (Fig. 2A and Supplementary Fig. 4A–C), an established marker of enteric neurons[23], consistent with previous estimates[24]. *Tph2* transcripts were observed in various subtypes of enteric neurons (Fig. 2A). *Tph2* transcripts were also present in sorted enteric neurons from the small intestines of adult mice but not in the intestinal epithelium or lamina propria immune cells (Fig. 2B). In contrast, *Tph1* transcripts were detected in the latter two small intestinal cell populations but not in enteric neurons (Fig. 2B).

To investigate the functions of enteric serotonergic neurons, we generated a conditional knockout mouse (*Tph2$^{fl/fl}$; Hand2-Cre*), leveraging the *Hand2* promoter (Fig. 2B), which is active in neural crest-derived cells including enteric neurons and other neurons in the PNS[25–27], but not in the brain[28]. We confirmed that *Tph2* transcripts were absent in the gut but preserved in the brain in *Tph2$^{fl/fl}$; Hand2-Cre* mice (Supplementary Fig. 4D), establishing that these animals are deficient in the expression of serotonin synthesis gene *Tph2*, at least in the enteric neurons. To further demonstrate the presence of enteric serotonergic neurons, we carried out whole-mount staining of cleared small intestine. Colocalization of serotonin and Tuj1, a neuronal cell marker, was observed within lamina propria neurons in *Tph2$^{fl/fl}$* but not in *Tph2$^{fl/fl}$; Hand2-Cre* animals (Fig. 2C and Supplementary movie 1). Moreover, previous research utilizing a radioenzymatic method[29] also reported the detection of enteric serotonergic neurons. The network of neuronal fibers in the myenteric plexus in *Tph2$^{fl/fl}$* and *Tph2$^{fl/fl}$; Hand2-Cre* mice appeared similar (Supplementary Fig. 4E), and comparable neuronal gene expression patterns were found in *Tph2$^{fl/fl}$* and *Tph2$^{fl/fl}$; Hand2-Cre* mice (Supplementary Fig. 4F). Together, these results suggest that serotonin produced by enteric neurons is not essential for the formation of the myenteric plexus and are consistent with studies of *Tph2$^{-/-}$* mice that showed the absence of *Tph2*-expressing neurons in the brain does not grossly influence brain development[30,31].

Like the *Tph2$^{-/-}$* mice, a reduction of fecal IgA (Fig. 2D) and small intestinal IgA$^+$ B cells was also observed in the *Tph2$^{fl/fl}$; Hand2-Cre* mice (Fig. 2E and Supplementary Fig. 4G). FACS analysis revealed that there were increased IgD$^+$ B cells and decreased IgA$^+$ B cells in the conditional KO animals (Fig. 2F and Supplementary Fig. 4H). The reduced abundance of IgA$^+$ B cells seemed to be restricted to the small intestinal lamina propria, as the number of IgA$^+$ B cells was similar in Peyer's Patches (PP) and mesenteric lymph nodes (MLN) from *Tph2$^{fl/fl}$* and *Tph2$^{fl/fl}$; Hand2-Cre* mice (Supplementary Fig. 4I). These data suggest that intestinal serotonergic neurons promote the development of IgA$^+$ B cells from IgD$^+$ precursors in the lamina propria of the small intestine.

Finally, we sought to investigate the role of intestinal serotonergic neurons in host defense. After oral infection with *Salmonella*, *Tph2$^{fl/fl}$; Hand2-Cre* mice had increased pathogen burdens in the spleen and liver (Fig. 2G) and reduced survival compared with co-housed littermate controls (Fig. 2H), suggesting that intestinal serotonergic neurons inhibit pathogen dissemination by augmenting the defense of intestinal barriers. Although direct comparison of the increased susceptibility to *Salmonella* in *Tph2$^{-/-}$* and *Tph2$^{fl/fl}$; Hand2-Cre* animals is not possible due to differences in their microbiota (Supplementary Fig. 3), the similarity in their phenotypes suggests that the absence of serotonin derived from intestinal neurons may largely account for the intestinal immune deficits observed in the whole-body *Tph2* KO animals.

## Serotonin remodels lamina propria DC function via the HTR7 receptor

To begin to assess how intestinal serotoninergic neurons modulate immune cell function, we analyzed the gene expression profiles of sorted small intestinal lamina propria (SiLP) immune cells (CD45$^+$) and found that HTR7 is the predominant serotonin receptor expressed in these cells (Fig. 3A). Re-analyses of single-cell RNA-seq data of SiLP immune cells[32] revealed that *Htr7* expression is primarily restricted to lamina propria dendritic cells (LPDC) (Fig. 3B), which was further confirmed by our FACS analysis of surface HTR7 expression in the SiLP-derived T cells, B cells and DC (Fig. 3C). Thus, this G-protein coupled receptor likely enables LPDC to respond to serotonin produced by serotonergic neurons. Consistent with this idea, the gene expression profiles of SiLP CD45$^+$ cells showed that *Tph2$^{-/-}$* mice exhibited lower expression of genes linked to DC differentiation compared to controls (Fig. 1C, D). Furthermore, FACS-based immune profiling revealed that both whole-body and conditional KO animals had a reduced abundance of LPDC compared to the respective littermate controls (Fig. 3D and Supplementary Fig. 5A).

To further characterize the effect of serotonin on LPDC, we treated FACS-isolated LPDC[33] in tissue culture with either 5-HT or a combination of serotonin and the serotonin receptor HTR7 antagonist SB269970 and analyzed their gene expression profiles. The abundance of 160 transcripts was increased >2-fold (adjusted *p*-value < 10$^{-5}$) in a HTR7-dependent manner (Fig. 3E). Many of the serotonin-induced, HTR7-dependent pathways were associated with immune regulation (Fig. 3F). Consistent with our findings of reduced B cell maturation and LPDC abundance in the *Tph2$^{fl/fl}$; Hand2-Cre* mice, these genes and pathways included loci linked to DC activation, such as *Cd83*[34] (Fig. 3E and Supplementary Fig. 5B), integrin signaling (Fig. 3F), which is associated with immune cell adhesion[35], as well as genes linked to DC-mediated activation and/or differentiation of B cells. The latter set of genes included *Cd40*[36], *Il6*[37], and *Lrrc32* (Fig. 3E and Supplementary Fig. 5C), a protein that promotes TGF-β activity[38,39] that is linked to stimulation of IgA class switching[40].

Co-culture experiments were used to test the hypothesis that serotonin-treated LPDC can promote the differentiation of IgD$^+$ B cells to IgA-secreting cells. Co-culture of IgD$^+$ B cells with LPDC triggered IgA production as described[37]; furthermore, the addition of serotonin augmented the abundance of IgA measured in supernatants (Fig. 3G).

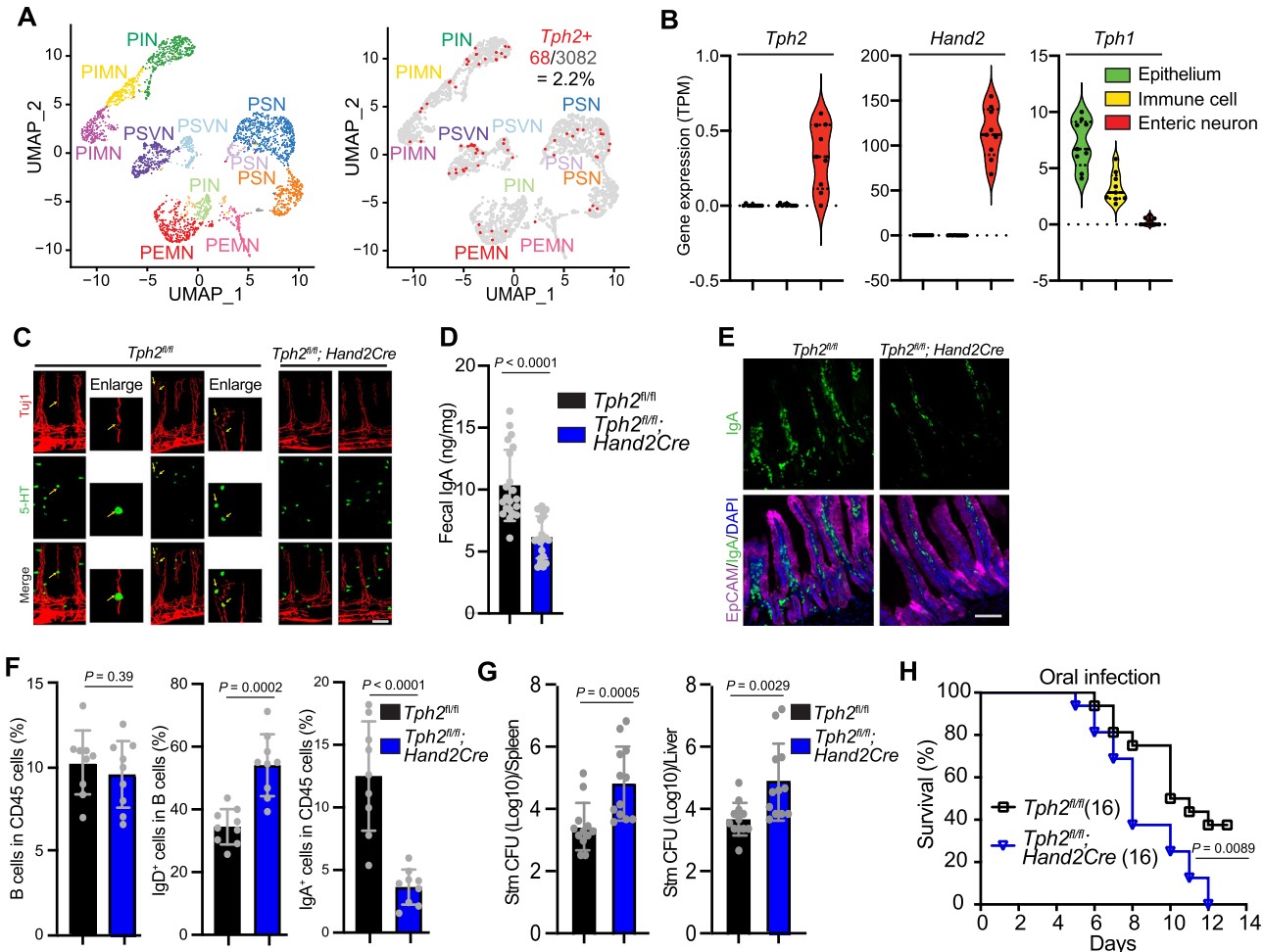

**Fig. 2 | Serotonin derived from enteric neurons contributes to the development of IgA B cells and mucosal defense in the intestine. A** scRNA-seq uniform manifold approximation and projection (UMAP) of *Tph2*⁺ enteric neurons from the small intestine of infant mice. Left, subtypes of enteric neurons according to ref. 61; PSN, putative sensory neurons; PIN, putative inhibitory neuron; PIMN, putative inhibitory motor neurons; PEMN, putative excitatory motor neurons; PSVN, putative secretomotor/vasodilator neurons were defined according to[61]. Right, distribution of *Tph2* transcripts among subsets of enteric neurons. **B** Detection of *Tph2*, *Hand2*, and *Tph1* in bulk RNA-seq analysis of the intestinal epithelium, lamina propria immune cells, and enteric neurons (*n* = 9 mice); TPM, Transcripts Per Million. **C** Representative whole-mount staining of small intestine from *Tph2*^fl/fl^ and *Tph2*^fl/fl^; *Hand2-Cre* mice. Neurons are detected with Tuj1 (red) and 5-HT positive cells with anti-serotonin antibody (green). Yellow arrows point to enteric serotonergic neurons (round-shaped cells) in the lamina propria, where serotonin co-

localizes with Tuj1. The triangular-shaped serotonin positive cells observed in both *Tph2*^fl/fl^ and *Tph2*^fl/fl^; *Hand2-Cre* mice likely represent enterochromaffin cells. Scale bar, 100 µm. Similar results were obtained in at least 5 animals. **D** Fecal IgA levels in *Tph2*^fl/fl^ and *Tph2*^fl/fl^; *Hand2-Cre* mice (*n* = 21). **E** Representative immunostaining of small intestine sections from *Tph2*^fl/fl^ and *Tph2*^fl/fl^; *Hand2-Cre* mice. Epithelial cells are detected with EpCAM (magenta), IgA cells with anti-IgA antibody (green), and nuclei with DAPI (blue). Scale bar, 100 µm. **F** Flow cytometry analyses of frequency of total B cells, IgD⁺ fraction of total B cells, and IgA⁺ cells in the lamina propria CD45⁺ cells from the small intestine of *Tph2*^fl/fl^ and *Tph2*^fl/fl^; *Hand2-Cre* mice (*n* = 9). **G** STm CFU burdens in spleens and livers of *Tph2*^fl/fl^ and *Tph2*^fl/fl^; *Hand2-Cre* mice 5 days post oral inoculation (*n* = 12). **H** Kaplan–Meier plot of survival of *Tph2*^fl/fl^ and *Tph2*^fl/fl^; *Hand2-Cre* mice following oral STm inoculation (*n* = 16). Data shown are means ± SD. Statistical analysis was performed by two-tailed Mann–Whitney test in **D**, **F**, **G**; and by a Log-rank test in **H**.

Serotonin stimulation of LPDC-mediated IgA production was ablated by an HTR7 antagonist, demonstrating that the effect of this neurotransmitter relies on HTR7-dependent signaling pathways. Together, these observations suggest that serotonin remodels LPDC function to promote IgA class switching.

The reduced abundance of LPDC in the *Tph2*^fl/fl^; *Hand2-Cre* mice could at least in part be due to reduced homing/retention of circulating DC in the intestine[41]. We used an in vivo DC homing assay (Fig. 3H), to test the hypothesis that enteric serotonergic neurons (ESN) promote DC homing/retention to the small intestine lamina propria as well as to PP and MLN. There was a greater abundance of labeled DC in the lamina propria of *Tph2*^fl/fl^ vs *Tph2*^fl/fl^; *Hand2-Cre* mice (Fig. 3I and Supplementary Fig. 5D), suggesting the presence of ESN in the lamina propria may promote the retention of DC in this site. In contrast, no difference in the recruitment of labeled bone-marrow derived DC to the MLN or PP of

*Tph2*^fl/fl^ vs *Tph2*^fl/fl^; *Hand2-Cre* mice (Supplementary Fig. 5E) was observed, indicating that the lymphoid tissue homing of DC is intact in the conditional KO mice. Furthermore, pretreatment of DC with an HTR7 antagonist prior to their intravenous injection reduced their abundance in the lamina propria (Fig. 3I and Supplementary Fig. 5D), suggesting that serotonin signaling through HTR7 on DC promotes the residency of these cells in the small intestine lamina propria. Since CCR7, which has been associated with DC homing to MLN and potentially to ectopic lymphoid tissue in the mucosa as well[42], was significantly induced by serotonin treatment of LPDC (Fig. 3E), we evaluated the role of this chemokine receptor in DC gut retention in the lamina propria. DCs derived from the bone marrow of CCR7-deficient animals had deficient homing and/or retention to the lamina propria of both WT and conditional KO mice, suggesting that this serotonin-induced receptor plays a crucial role in DC gut homing and/or retention

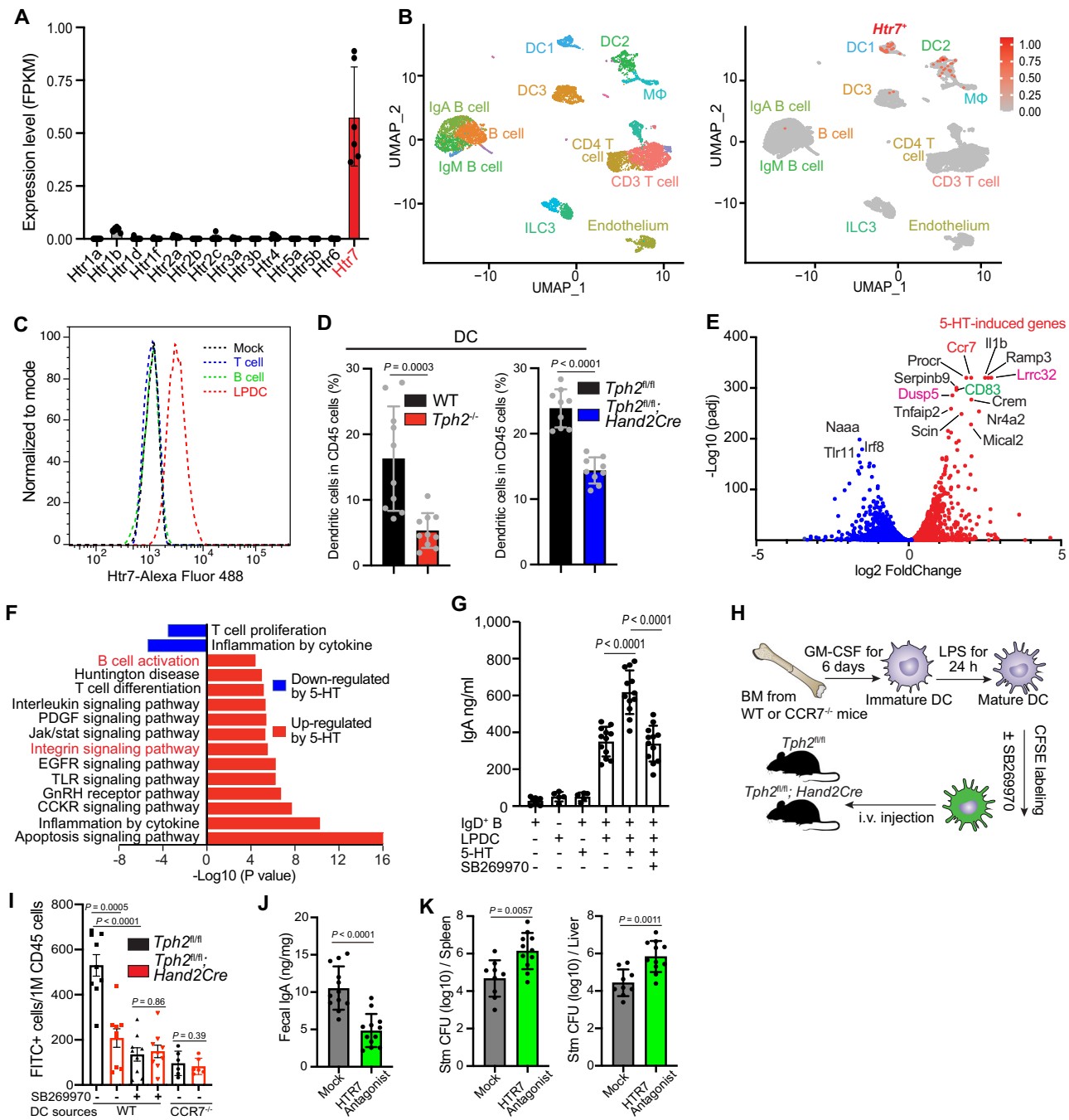

to the lamina propria. This mechanism could account for the reduced abundance of LPDC in mice deficient in *Tph2* in ESN. Together, these observations suggest a new axis of interaction between ESN and the immune system: serotonin derived from enteric neurons promotes both retention and remodeling of DC function through HTR7 signaling, which in turn could induce IgA⁺ B cell development.

We also measured the effect of the HTR7 antagonist on IgA B cell differentiation and *Salmonella* infection, to use a non-genetic approach to probe the importance of serotonin-HTR7 interactions in vivo. We found that animals treated with the HTR7 antagonist phenocopied the conditional *Tph2* KO mice in that they exhibited decreased levels of fecal IgA (Fig. 3J) and increased susceptibility to *Salmonella* infection (Fig. 3K). Although the HTR7 antagonist could also block the effects of serotonin produced by cells other than neurons, these findings provide additional support for the role of the HTR7 signaling pathway in the pDC-IgA-*Salmonella* susceptibility circuitry.

## Intestinal serotonergic neurons control the abundance and function of lamina propria pDC

We hypothesized that in vivo ESN-derived serotonin remodels LPDC function in an HTR7-dependent manner. To test this idea, LPDC from *Tph2*^fl/fl and *Tph2*^fl/fl; *Hand2-Cre* mice were isolated by positive selection for single-cell RNA-seq (Supplementary Fig. 6A). While the fraction of MHCII⁺ cells was similar in the *Tph2*^fl/fl; *Hand2-Cre* mice and WT controls, there was a lower proportion of LPDC in the conditional KO animals (Supplementary Fig. 6B–D), as observed above by FACS-based analysis (Fig. 3D). Seven clusters of CD11c⁺ cells were identified (Fig. 4A), including plasmacytoid dendritic cell (pDC), conventional DC 1 (cDC1) and conventional DC type 2 (cDC2) based on marker gene expression[43](Fig. 4B). Both pDC and cDC1 were found to express *Htr7*, whereas cDC2 cells expressed very low levels of this serotonin receptor (Fig. 4C). Among LPDC, the proportion of pDC was reduced in *Tph2*^fl/fl; *Hand2-Cre* mice vs *Tph2*^fl/fl animals (44% to 25%), but the

**Fig. 3 | Serotonin promotes retention and remodeling of DC function through HTR7 signaling and stimulates DC-dependent IgA B cell development. A** HTR7 is the dominant serotonin receptor in lamina propria CD45⁺ cells. FPKM (Fragments Per Kilobase of transcript per Million fragments mapped) values of all serotonin receptors in lamina propria CD45⁺ cells. *n* = 6 mice. **B** UMAP plot of single-cell transcriptome data[32] of small intestinal lamina propria immune cells showing that expression of the *Htr7* gene is restricted to LPDC cells. **C** Flow cytometry analyses of HTR7 expression in small intestinal lamina propria T cells, B cells, and DC. **D** Flow cytometry analyses of frequency of DC (Live, CD45⁺, MHC-II⁺, CD64⁻, CD11c⁺) in lamina propria CD45⁺ cells from the small intestine of mice with the indicated genotypes (*n* = 10 in WT and *Tph2⁻/⁻* group, *n* = 9 in Cre+/−; *Tph2^fl/fl* group). **E** Differential gene expression in LPDCs treated with serotonin ±HTR7 antagonist SB269970. DESeq2 was used to calculate fold-change and adjusted *p*-value. Red points represent genes that exhibited significantly increased expression in the serotonin treatment group compared to the group treated with serotonin along with the HTR7 antagonist, while blue points indicate genes that showed significantly decreased expression in the serotonin treatment group compared to the group treated with serotonin along with the HTR7 antagonist. **F** Gene Ontology pathway analysis of data from **E**. **G** Tissue culture assay of IgA production from IgD⁺

B cells that were cocultured with LPDC, serotonin, and a HTR7 antagonist (SB269970) as indicated. The levels of IgA in the supernatants were measured by ELISA after 6 days of coculture (*n* = 12). Data are representative of three independent experiments. **H** Schematic of the system used to analyze BMDC homing to the gut in *Tph2^fl/fl* and *Tph2^fl/fl*; *Hand2-Cre* mice. Equal numbers of CFSE-labeled mature WT or CCR7⁻/⁻ BMDC were transplanted into *Tph2^fl/fl* and *Tph2^fl/fl*; *Hand2-Cre* mice and the CFSE⁺ cells in different tissues were analyzed 18 h post-transplantation. **I** Frequency of CFSE⁺ BMDC among CD45⁺ cells in the lamina propria of *Tph2^fl/fl* and *Tph2^fl/fl*; *Hand2-Cre* mice. HTR7 antagonist SB269970 was used to inhibit the HTR7 signaling pathway (*n* = 9), while CCR7⁻/⁻ cells were employed to block CCR7-dependent signaling (*n* = 6). **J** Fecal IgA levels in mock and HTR7 antagonist-treated mice. The mice were treated with either i.p PBS or an HTR7 antagonist every other day for 10 days. Fecal IgA measurements were carried out after the 10-day HTR7 antagonist treatment. *n* = 12 mice per group. **K** STm CFU burdens in spleens and livers of mock and HTR7 antagonist-treated mice 5 days post oral inoculation. The mice were treated as described in **J** before oral *Salmonella* infection. *n* = 9 mice in the mock and *n* = 11 mice in the treated group. Data shown are means ± SD. Statistical analysis was performed by a two-tailed Mann–Whitney test in **D**, **G**, and **I**–**K**.

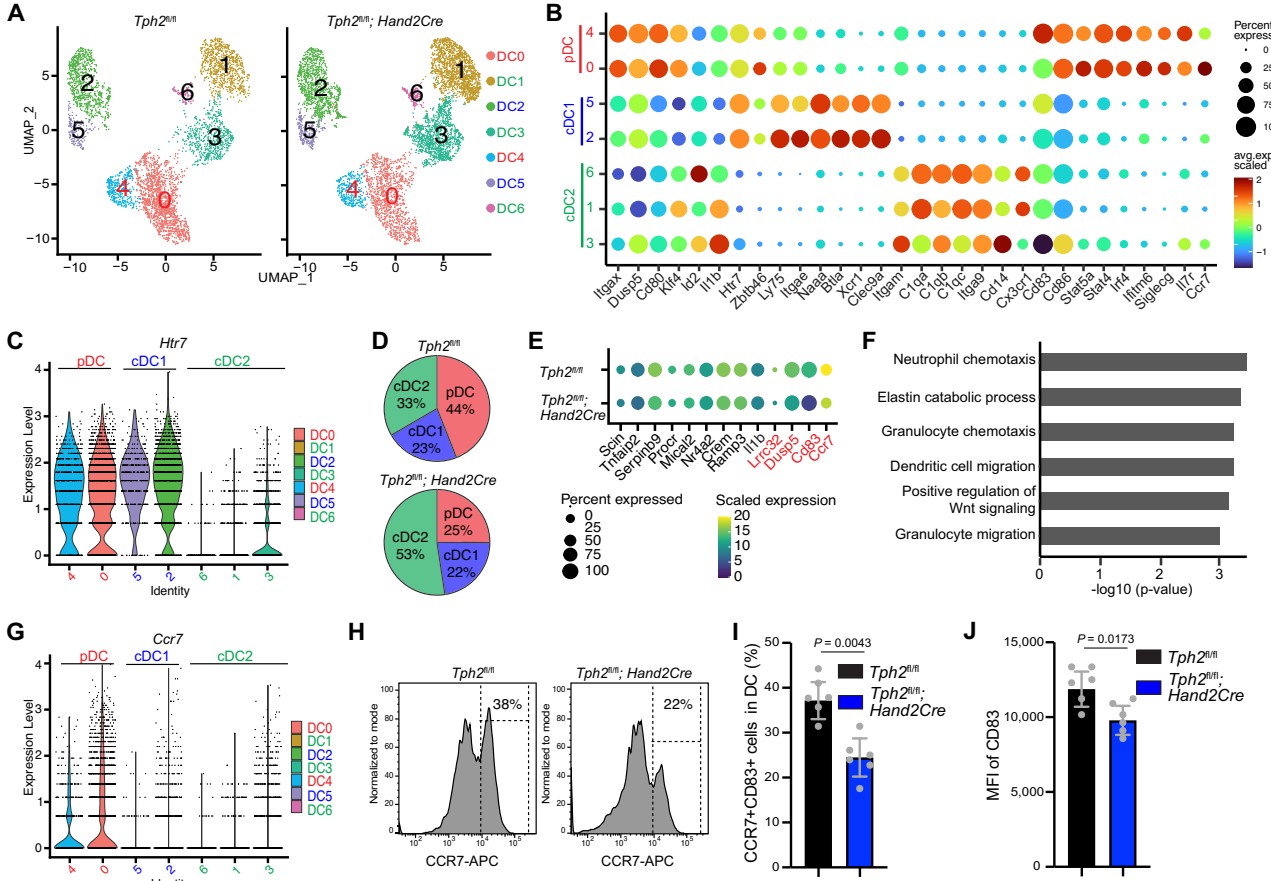

**Fig. 4 | Enteric serotonergic neurons remodel pDC function in vivo. A** UMAP plot of single-cell transcriptome data of small intestinal LPDC from *Tph2^fl/fl* and *Tph2^fl/fl*; *Hand2-Cre* mice. **B** Marker gene expression used for annotation of LPDC subsets. **C** Violin plots depicting the range of expression levels of *Htr7* in distinct LPDC clusters. **D** Pie charts showing the proportion of pDC is decreased in *Tph2^fl/fl*; *Hand2-Cre* vs *Tph2^fl/fl* mice. **E** Comparison of the expression of the indicated genes in pDC between *Tph2^fl/fl* and *Tph2^fl/fl*; *Hand2-Cre* mice. The dot color reflects the level of gene expression, and the dot size represents the percentage of cells expressing the indicated gene. The genes labeled in red exhibit decreased expression in pDC from *Tph2^fl/fl*; *Hand2-Cre* mice and are consistent with the serotonin-induced genes observed in vitro. **F** GO Pathway analysis of genes that exhibited decreased

expression in pDCs of *Tph2^fl/fl*; *Hand2-Cre* vs *Tph2^fl/fl* mice. **G** Violin plots depicting the range of expression levels of *Ccr7* in distinct LPDC clusters. **H** Representative histogram plots showing the percentage of pDC (Live, CD45⁺, MHC-II⁺, CD64⁻, CD11c⁺, CCR7⁺) in LPDC from the small intestine of *Tph2^fl/fl* and *Tph2^fl/fl*; *Hand2-Cre* mice by flow cytometry. **I** Flow cytometry analyses of frequency of pDC in LPDC from the small intestine of *Tph2^fl/fl* and *Tph2^fl/fl*; *Hand2-Cre* mice. *n* = 6 mice per group. **J** Flow cytometry analyses of CD83 expression on pDC from the small intestine of *Tph2^fl/fl* and *Tph2^fl/fl*; *Hand2-Cre* mice. MFI mean fluorescence intensity. *n* = 6 mice per group. Data shown are means ± SD. Statistical analysis was performed by a two-tailed Mann–Whitney test in **I** and **J**.

fraction of cDC1 was similar in the two groups (Fig. 4D). Together, these findings suggest that interactions between 5-HT produced by ESN and the HTR7 serotonin receptor on pDC promote the homing to and/or survival of pDC in the lamina propria of the small intestine.

Additional comparisons of the sc-RNA-seq data showed that some of the most highly serotonin-induced genes in LPDC in vitro (Fig. 3E) exhibited reduced expression in pDC in vivo in *Tph2^fl/fl^; Hand2-Cre* mice vs control animals; these genes included TGF-β activator *Lrrc32*, DC activation marker *Cd83*, *Dusp5* as well as *Ccr7* (Fig. 4E). These comparisons of gene expression in pDC from *Tph2^fl/fl^; Hand2-Cre* vs *Tph2^fl/fl^* mice also uncovered reduced expression of genes implicated in chemotaxis and migration, further buttressing the idea that ESN play critical roles in pDC homing and/or retention (Fig. 4F). Among LPDC, only pDC expressed *Ccr7* (Fig. 4G), a chemokine receptor that plays a critical role in cell-cell interaction[44,45]. Thus, the reduced homing/retention of pDC in the lamina propria may, in part, be attributable to reduced *Ccr7* expression in these cells (Figs. 3E, I, 4E, F). CCR7 is also a useful cell surface marker for pDC[46] and FACS analysis of CCR7+ LPDC confirmed the reduction in the abundance of pDC in the *Tph2^fl/fl^; Hand2-Cre* mice compared to the control animals (Fig. 4H, I and Supplementary Fig. 6E–G). Similar FACS analysis confirmed the reduction of the pDC activation marker CD83 in the *Tph2^fl/fl^; Hand2-Cre* mice (Fig. 4J), providing further verification of the scRNA-seq results. Collectively, these observations are consistent with the idea that serotonin derived from ESN stimulates the expression of genes linked to the homing/retention and differentiation of a specific subset of LPDC, plasmacytoid dendritic cells, supporting the concept that there are functionally important interactions between ESN and pDC in vivo.

### Bidirectional communication between enteric neurons and LPDC

Our understanding of how interactions between the ENS and the gut innate immune system contribute to intestinal homeostasis is limited. Whole-mount immunofluorescence imaging of small intestinal villi was used to first address whether DC are found in close proximity to enteric neurons. Neuronal projections ramifying within the lamina propria were often observed in close proximity to LPDC, suggesting that interactions between these cell types are plausible (Fig. 5A).

We used CellChat[47] to infer and analyze the potential intercellular communication networks within the ENS-innate immune axis. For these analyses, the scRNA-seq data from the 4347 enteric neurons and 3284 LPDC sequenced in this study were utilized (Fig. 5B and Supplementary Fig. 7A). While most of the cell-cell interaction pathways detected in these analyses were between neuron subtypes, predicted interactions between neurons and DCs and between DC subtypes were also found (Fig. 5C). Generally, enteric neurons and LPDCs appeared to rely on distinct pathways for communication (Supplementary Fig. 7B).

Among the ligand-receptor interactions between enteric neurons and DCs respectively inferred by these analyses (Fig. 5D, G), we uncovered significant interactions driven by neuronal production of 'immune' ligands such as *Il7*, *Flt3l* and *Ccl25* and expressed DC receptors, *Il7R*, *Flt3*, and *Ccr9*, that are known to regulate DC function[48] (Fig. 5D, E). Interactions between neuronal *Ptn*, a ligand linked to growth and cytokine regulation[49], and *Ncl* and *Sdc4* receptors on LPDC were the most significant interaction detected (Fig. 5E, F and Supplementary Fig. 7C), suggesting that enteric neurons play a role in controlling proliferation/activation of LPDC. Notably, these analyses independently predicted the ENS-DC interaction through the serotonin-HTR7 pathway (Fig. 5D, E).

LPDC produced ligands, including several chemokines were predicted to interact with chemokine receptors, such as *Ackr1*, on enteric neurons (Fig. 5G, H), suggesting that innate immune cells regulate ENS function. LPDC also appear to control neuronal energy homeostasis, since the Nampt/Visfatin-Insulin receptor pathway was the most significant predicted interaction (Fig. 5H, I and Supplementary Fig. 7D).

Finally, these analyses identified putative interactions between specific neuron and LPDC clusters. For example, *Il7* expression was restricted to enteric neuron cluster 2, putative sensory neurons, while IL7 receptors expression was exclusive to pDC (Fig. 5J–L), suggesting that enteric sensory neurons may control the development of this DC subset[50]. Although additional studies to validate this interaction and to elucidate its significance are necessary, this computational approach strongly suggests that there is a larger range of bidirectional crosstalk between enteric neurons and innate immune cells than was appreciated.

## Discussion

Here, we uncovered a new role of neuronal serotonin in immune homeostasis at the intestinal mucosal barrier. Our findings suggest that serotonergic neuronal circuits, which are thought to function in regulation of emotion, also play a role in governing intestinal innate immune defense. Mechanistic studies showed that serotonin produced by neurons in the intestine targets the HTR7 receptor on LPDC. Interactions between 5-HT and HTR7 appear to promote the homing/retention and differentiation of a specific subset of DC, pDC, which are known to promote B cell class switching[51]. Thus, deficiencies in pDC abundance and function in the absence of intestinal neuronal serotonin signaling likely account for the reduction in IgA B cells and fecal IgA observed in the *Tph2^fl/fl^; Hand2-Cre* mice. Deficiencies in pDC and reduced IgA could both contribute to the increased susceptibility of the conditional KO mice to oral infection with *Salmonella*; deficiencies in other cell types could also impact susceptibility to *Salmonella*. Thus, these studies begin to uncover molecular pathways by which serotonergic neuronal circuits, which have been linked to the control of emotion, also impact immune function.

Whole-body and conditional *Tph2* KO animals exhibit decreased gut motility[52,53]. It is possible that reduced gut motility could potentially contribute to the increased *Salmonella* burden observed in the livers and spleens of the *Tph2* deficient animals. Interestingly, we found that *Salmonella* infection increased intestinal motility in WT mice and that *Tph2* KO animals also exhibited a similar increase in gut motility with *Salmonella* infection (Supplementary Fig. 8A). However, even in the setting of infection, the *Tph2* KO animals had reduced gut motility compared to the WT mice. Thus, the gut motility defect observed in *Tph2* KO animals could contribute to their susceptibility to *Salmonella*. To further investigate whether changes in gut motility account for increased *Salmonella* burdens in distal organs, we measured the effect of an HTR7 antagonist on gut motility. Mice treated with the HTR7 antagonist did not display reduced gut motility (Supplementary Fig. 8B) but showed increased susceptibility to *Salmonella* infection (Fig. 3K), suggesting that gut motility is not the only factor that explains the elevated susceptibility to *Salmonella* infection in *Tph2* KO animals.

Recent research has demonstrated that enteric neurons play roles in regulating various immune cell types, including macrophages, innate lymphocyte cells, and Treg cells[14,54]. Our findings illuminate a novel gut neuro-immune interaction in which serotonin, a neurotransmitter, controls LPDC function to promote IgA B cell differentiation. The HTR7 on lamina propria pDC appears to be critical for ESN to activate pDCs, suggesting that serotonin derived from enteric serotonergic neurons may account for some of the diverse immune phenotypes linked to HTR7[55]. Notably, human scRNA-seq studies have revealed that, like mice, HTR7 is the only serotonin receptor expressed on intestinal DCs (Supplementary Fig. 8C)[56,57]. Thus, our findings, along with other studies[58,59], raise the possibility that drugs that activate or inhibit HTR7 may be useful as immunomodulatory agents for treating human enteric infections as well as for managing inflammatory bowel disease. Furthermore, our findings may explain some of the observed adverse outcomes in patients with *Tph2* gene variants, including susceptibility to viral infection[16].

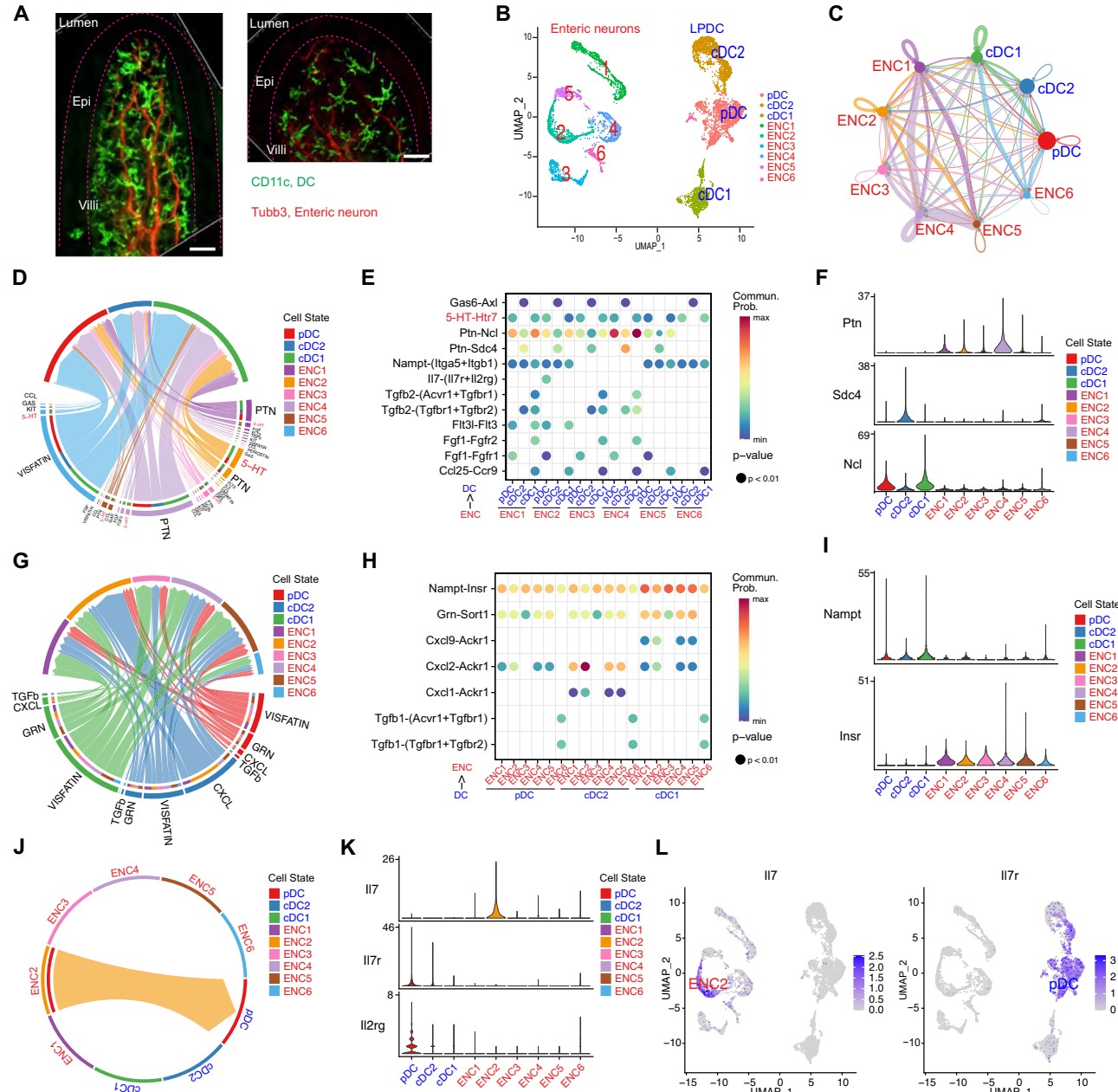

**Fig. 5 | Bidirectional communication between enteric neurons and LPDC.**
**A** Representative whole-mount immunostaining of small intestine villi from CD11c-DTR/GFP mice. LPDC are marked with CD11c (green), and neuronal projections are labeled with anti-Tubb3 antibody (Red). Scale bar, 20 μm. Similar results were obtained in at least 5 animals. **B** UMAP plot of single-cell transcriptome data of enteric neurons and LPDC generated in this study that was used for analyzing cell-cell communication with CellChat. **C** Circle network diagram of significant cell-cell interaction pathways. Arrows and line color indicate direction (ligand: receptor) and line thickness indicates the strength of the predicted interaction. The round loops represent the interactions within the same cell type. **D–F** Enteric neurons to LPDC communication pathways. **D** Chord diagram showing all the significant interactions from enteric neurons to LPDC. **E** Dot plot showing the key ligand-receptor interactions in inferred communication pathways between subsets of enteric neurons and LPDC. Empty spaces indicate that the communication probability is zero. *p*-values were computed with a one-sided permutation test. The color scale represents communication probability as calculated in CellChat. **F** Violin plots illustrating the gene expression levels of the PTN signaling pathway, with ligand

expression in enteric neuron clusters and receptor expression in LPDC clusters. **G–I** LPDC to enteric neurons communication pathways. **G** Chord diagram showing all the significant interactions from LPDC to enteric neurons. **H** Dot plot showing the key ligand-receptor interactions in inferred communication pathways between subsets of LPDC and enteric neurons. Empty spaces indicate that the communication probability is zero. *p*-values were computed with a one-sided permutation test. The color scale represents communication probability as calculated in CellChat. **I** Violin plots illustrating the gene expression levels of the Nampt-Insr signaling pathway, with ligand expression in LPDC clusters and receptor expression in enteric neuron clusters. **J** Chord diagram displaying the communication network between putative enteric sensory neurons and pDC, highlighting the targeting of *Il7* from enteric sensory neurons to the *Il7r* located on the pDC. **K** Violin plots illustrating the gene expression levels of the IL7 signaling pathway, with ligand expression in enteric sensory neurons and receptor (a heterodimer of *Il7r* and *Il2rg*) expression in pDC. **L** UMAP plot derived from single-cell transcriptome data showing expression of *Il7* in enteric sensory neurons (left) and Il7 receptor in pDC (right).

Additional studies are required to decipher whether signaling between ESN and pDC is through a synapse-like mechanism, where there is a direct interaction between pDC and ESN, or via a less direct pathway that ultimately promotes pDC retention and activation in the small intestinal lamina propria (Supplementary Fig. 8D). Whole mount immunofluorescence imaging revealed that there is close apposition of intestinal neurons and DC, suggesting that a direct ESN-pDC interaction is possible. Regardless of the precise mechanism through which serotonergic neurons in the intestine control pDC function, the absence of *Tph2*-derived serotonin in the intestine impaired mucosal defense against *Salmonella*, demonstrating the importance of these neuronal circuits in host defense. Furthermore, it will be intriguing to investigate whether serotonin derived from enterochromaffin cells affects DC and B cell function.

Gene expression profiling uncovered the cell-type specificity of the expression of serotonin receptors. Enteric sensory neurons expressed *Htr3a* (Supplementary Fig. 7E, F), whereas *Htr4* was expressed in intestinal epithelial cells (Supplementary Fig. 7G, H), and *Htr7* in LPDC (Fig. 3A–C). Although further investigation is warranted, it may be possible to develop drugs that specifically bind these receptors, enabling precise targeting of specific cell types in the intestine. Engineering these compounds so that they are unable to penetrate the blood-brain barrier could prevent untoward central nervous system-related side effects.

Bioinformatic analyses using CellChat and similar pipelines[60] to systematically and quantitatively infer intercellular communication networks will be invaluable for deciphering neuro-immune interactions. The predicted interactions between enteric neurons and innate immune cells that we uncovered suggest that there is a complex bidirectional dialogue between specific subsets of enteric neurons and DC. Functional analyses based on these inferences will reveal new insights into the roles of the enteric nervous system in innate immune regulation, as well as the potential remodeling of enteric neuronal function by innate immune cells.

## Methods
### Mice
C57BL/6 (Strain #:000664), Tg(*Uchl1*-HISTH2BE/mCherry/EGFP*) Fsout/J (Strain #:016981), CD11c-DTR/GFP (Strain #:004509), *Sert*[−/−] (Strain #:008355), *Ccr7*[−/−] (Strain #:006621) and *Tph2*[flox/flox] mice (Strain #:027590) were purchased from the Jackson Laboratory (Bar Harbor, ME, USA); *Tph2*[−/−] mice were a generous gift from Dr. Gerard Karsenty (Columbia University, NY, USA); *Hand2-Cre* transgenic mice were a generous gift from Dr. Ruaidhrí Jackson (Harvard Medical School, MA, USA). *Tph2*[flox/flox] mice were backcrossed to C57BL/6J background for at least six generations. All the experimental and control animals were co-housed, maintained on a 12-hour light/dark cycle and a standard chow diet at the Harvard Institute of Medicine specific pathogen-free (SPF) animal facility. The mice were euthanized by cervical dislocation following an overdose of isoflurane inhalation. Both male and female mice, 6–8 weeks old, were used unless otherwise specified in the figure legend. Animal experiments were performed according to guidelines from the Center for Animal Resources and Comparative Medicine at Harvard Medical School. All protocols and experimental plans were approved by the Brigham and Women's Hospital Institutional Animal Care and Use Committee (Protocol #2016N000416).

### Isolation and purification of nuclei from enteric neurons
The isolation of enteric neurons from Tg(*Uchl1*-HISTH2BE/mCherry/EGFP*) Fsout/J mice was performed as previously described[61]. Briefly, fresh-frozen intestinal tissues were disaggregated in 1 mL of CST buffer (0.49% (w/v) CHAPS (220201, EMD Millipore), 146 mM NaCl (S6546, Sigma), 1 mM CaCl₂ (97062-820, VWR), 21 mM MgCl₂ (M1028, Sigma), 10 mM Tris pH 8.0 (AM9855G, Thermo Fisher) with mild chopping using scissors for 10 minutes on ice. Large debris was removed with a 40 μm strainer (Falcon). The isolated nuclei were stained with 4′,6-diamidino-2-phenylindole (DAPI, D1306, Thermo Fisher Scientific) and sorted as DAPI⁺, mCherry⁺ nuclei using an SH800 Cell Sorter (Sony Biotech). The purified nuclei were used for single-nucleus RNA-seq or bulk RNA-seq analysis.

### Intestinal epithelium and immune cell isolation and purification
Colonic and small intestinal tissues were dissected, and Peyer's patches were discarded. The epithelium was isolated by 250 g stirring at 37 °C in RPMI medium (61870036, Gibco) containing 5 mM EDTA (AM9260G, Invitrogen), 1 mM dithiothreitol (10197777001, Sigma) and 2% (vol/vol) FBS (10082147, Gibco) for 15 min. The isolated epithelium was used for bulk RNA-seq analysis.

To isolate lamina propria lymphocytes, the epithelium-depleted intestinal tissues were washed in RPMI medium with 5% (vol/vol) FBS, further minced into small pieces, and then digested by 250 rpm stirring at 37 °C in RPMI medium containing collagenase D (0.5 mg/ml, 11088866001, Sigma), Dispase II (0.5 mg/ml, 17105041, Gibco), DNase I (50 ug/ml, 10104159001, Roche) and 5% (vol/vol) FBS for 40 min. The digested tissues were filtered, and lamina propria cells were collected by centrifugation. The pellets were resuspended, and the lamina propria lymphocytes were isolated by Percoll (40%/80%, GE17-0891-01, GE Healthcare) gradient centrifugation. Mesenteric lymph nodes and Peyer's patches were mechanically disrupted. The purified immune cells were used for FACS analysis and sorting. The sorted CD45⁺ cells were used for bulk RNA-seq analysis.

### FACS analysis and cell sorting
All the antibodies used in this study are listed in Supplementary Table 1. Single-cell suspensions were blocked with an antibody against CD16/32 (2.4G2, Becton Dickinson) and then stained with LIVE/DEAD™ fixable aqua cell stain kit (L34957, Invitrogen), and antibodies against CD45 (30-F11), MHC-II (M5/114.15.2), CD64 (X54-5/7.1), CD11c (HL3), CCR7 (4B12), CD83 (Michel-19), B220 (RA3-6B2), IgD (11-26c), and IgA (C10-3) all from Becton Dickson. Anti-HTR7 polyclonal antibody is from MyBiosource, stained cells were analyzed with a FACSymphony analyzer (Becton Dickson) or sorted on an SH800 Cell Sorter; analyses were performed with FlowJo software (v10.8).

### Single nucleus RNA-Seq
The 10× Genomics protocol was performed on a single sorted enteric nucleus as described[61]. Libraries were generated using Chromium Next GEM Single Cell 3′ Reagent Kits v3.1 (10×) according to the manufacturer's protocol; library quality was assessed by the Tapestation High Sensitivity D5000 ScreenTape (Agilent) and sequenced using a NextSeq 500 sequencer (Illumina). Data analysis was performed using the 10× Genomics Cloud Analysis Cell Ranger pipeline and Seurat v4.3.0 with the SCT normalization method. Low-quality cells were removed by the parameters nfeature_RNA > 500, nCount_RNA > 2000 & nCount_RNA < 50000, and percent.mt <5. To generate UMAPs, we integrated our data with published small intestine data in Drokhlyansky et al.[61] and performed a principal component analysis (PCA) using RunPCA function (npcs = 50), and then cell clusters were identified using FindNeighbors (dims = 1:15) and FindClusters (resolution = 0.4) functions. RunUMAP function (dims = 1:15, n.neighbors = 15, min.dist = 0.3, spread = 1) was used to create UMAP plots. The cell-type assignment was performed using the clustifyr R package (v1.8.0, clustifyr[62]) based on the cell-types presented in Drokhlyansky et al.[61].

### Single-cell RNA-Seq of LPDC
Lamina propria lymphocytes were purified on percoll gradients from *Tph2*[fl/fl] and *Tph2*[fl/fl]; *Hand2Cre* mice and then DC were isolated using the EasySep™ Mouse CD11c Positive Selection Kit II (18781, StemCell). Library generation, sequencing, and data analysis were performed as described above. MHCII-negative cell clusters were computationally

removed prior to analyses of MHCII⁺ cell clusters. LPDC cell clusters specific analyses were carried out after the computational removal of CD11c negative cell clusters.

## Bulk RNA-seq and data analysis

Purified epithelial and immune cells were lysed in Trizol (15596018, Invitrogen), and RNA was extracted with the RNeasy mini Kit (74106, Qiagen) according to the manufacturer's instructions. RNA-seq libraries were prepared using the KAPA mRNA HyperPrep kit (50-196-5308, Roche) or NEBNext Multiplex Oligos for Illumina kit (E7335L, NEB). Libraries were analyzed using a High Sensitivity D1000 Screen-Tape (Agilent) and sequenced on a NextSeq 550 or 1000/2000 (Illumina). For data analysis, reads were trimmed using Trim Galore with automatic adapter sequence detection. Then, trimmed reads were mapped to the mouse reference genome (mm10) and annotated using STAR v2.7.3a with default parameters. The number of mapped reads to each gene was counted by featureCounts of the Subread package using mouse GENCODE annotation M25 (GRCm38.p6). Transcripts Per Kilobase Million (TPM) were calculated by dividing the number of read counts by the length of the gene in kilobases to yield reads per kilobase. Fragments Per Kilobase of transcript per Million fragments mapped (FPKM) were also calculated as described[63]. Differentially expressed genes were identified using the R-package DESeq2 (v1.36.0) with an adjusted $p$-value < 0.05.

## Histology, whole-mount staining and tissue immunofluorescence

For immunofluorescence analysis, samples from the small intestine were collected and flushed with PBS and fixed in 4% PFA followed by washing with PBS. Tissue samples were then soaked in 30% sucrose and embedded in Optimal Cutting Temperature Compound (23-730-571, Fisher Scientific) and stored at −80 °C before sectioning on a CM1860 UV cryostat (Leica). 8 µm-thick slides were stained with FITC-anti IgA (C10-3, Becton Dickson) and Percp/cy5.5 anti-EpCAM (G8.8, Biolegend) antibodies at 4 °C overnight in PBS. Nuclei were stained with DAPI at RT for 5 min in the dark. For whole-mount staining of villi, a small segment of the proximal small intestine of CD11c-DTR/GFP mice was removed and placed in ice-cold PBS and then transferred to a Sylgard (Dow-Corning)-coated Petri dish. The intestine was opened along the mesenteric border and pinned flat, the tissue was stained overnight with Alexa Fluor 568 anti-Tubulin β3 antibody (TUJ1, Biolegend) as well as Alexa Fluor 488 anti-GFP antibody and rinsed in PBS before imaging. For staining of myenteric plexuses, the intestine was opened along the mesenteric border and pinned flat, and then the mucosa and submucosa were removed with forceps to expose the muscularis propria layer. The resulting muscularis propria preparation was stained overnight with Alexa Fluor 647 anti-Tubulin β3 antibody (TUJ1, Biolegend) and rinsed in PBS before imaging with an Eclipse Ti confocal microscope with a 20× objective (Nikon).

For whole-mount staining, freshly dissected small intestine was fixed with 4% PFA overnight on ice. The tissues were bleached by 6% hydrogen peroxide in methanol in 4 °C for 1 h and blocked by 1% donkey serum in PBSGT (0.5%Triton X-100, 0.2% Gelatin in PBS) for 2 days with shaking in a 37 °C incubator. After incubating with primary antibodies against Tuj1 and 5-HT in PBSGT for 1 day in the dark at 37 °C incubator with shaking, samples were washed and cleared by Dichloromethane and Ethyl. Tissues were then laid on slides and sealed under a coverslip. Z stack images were acquired on an Olympus Stella Confocal microscope.

## ELISA for fecal IgA

Fresh mouse feces were homogenized in 10× volume (v/w) of PBS containing the protease inhibitor cocktail Complete EDTA-free (11836153001, Roche), using a bead-beater (Biospec, Bartlesville, OK, USA) for 20 s, and then centrifuged at 5000 g for 15 min at 4 °C.

Supernatants were further spun down at 15,000 g for 5 min at 4 °C, and the IgA level in the final supernatant was determined using the Mouse IgA ELISA kit (88-50450-86, Thermo Fisher Scientific) according to the manufacturer's protocol.

## RT-qPCR quantification of gene expression

The brainstem and small intestine muscularis propria were isolated and lysed in Trizol (15596018, Invitrogen). RNA was purified with the RNeasy mini Kit (74106, Qiagen) according to the manufacturer's instructions. RNA samples were treated for residual DNA contamination using Ambion Turbo DNA-free DNase (AM1907, Invitrogen). Purified RNA was reverse transcribed for quantitative RT-PCR (RT-qPCR) by adding 10 µg of total RNA to a mixture containing oligo dT (N8080128, Thermo Fisher Scientific), 25 mM dNTPs (R0191, Thermo Fisher Scientific), 0.01 M dithiothreitol, reaction buffer and 200 units of SuperScript III reverse transcriptase (18080085, Invitrogen). cDNA was diluted 1:50 in dH2O and mixed with an equal volume of target-specific primers and Roche SYBR master mix (04707516001, Roche) or TaqMan™ Gene Expression Master Mix (4369016, Applied Biosystems). Primer pairs were designed to minimize secondary structures and a melting temperature of 60 °C using the primer design software Primer 3. Primer sequences are listed in Supplementary Table 2. For data normalization, quadruplicate Ct values for each sample were averaged and normalized to Ct values of the control gene *Gapdh*. The relative gene expression level was determined by normalizing expression in the WT.

## 16S rRNA profiling of the gut microbiota

Fecal DNA was extracted from mouse fecal pellets with the QIAGEN QIAamp Fast DNA Stool Mini Kit (51604, Qiagen) according to the manufacturer's instructions. Purified DNA samples were amplified with barcoded primer pairs: 341F and 805R. The amount of PCR products were quantified with the Qubit kit (Q32854, Invitrogen) and then the same amount of DNA from each sample was pooled for sequencing on MiSeq (Illumina). Raw sequencing data were analyzed by QIIME2[64]. In brief, the data were imported into QIIME2 and demultiplexed and the DADA2 pipeline was used for sequencing quality control, and a feature table was constructed. The feature table was used for alpha and beta diversity analysis as well as for taxonomic analysis and differential abundance testing.

## BMDC gut homing assay

Bone marrow-derived dendritic cells (BMDCs) were generated from a flushed bone marrow suspension from WT or *Ccr7*⁻/⁻ mice as previously described[65]. In brief, $5 \times 10^6$ bone marrow cells from the femurs of C57BL/6J mice were cultured in 10 ml medium containing RPMI 1640 (11875093, Gibco) with 10% FBS (10082147, Gibco), β-Mercaptoethanol (21985023, Gibco), GlutaMAX (35050061, Gibco) and penicillin-streptomycin (10378016, Gibco) supplemented with 100 ng/ml granulocyte-macrophage colony-stimulating factor (GM-CSF; 315-03, PeproTech). Another 10 ml of GM-CSF-supplemented medium was added every 3 days. To generate mature BMDCs, 1 µg/ml lipopoly-saccharide (LPS; LPS25, Sigma) was added to the cell suspensions on day 8 for 24 h. At day 9, mature BMDCs were harvested, and labeled with CellTrace CFSE (C34554, Thermo Fisher Scientific) and then injected intravenously into *Tph2ᶠˡ/ᶠˡ* and *Tph2ᶠˡ/ᶠˡ; Hand2-Cre* mice. Recipient mice were euthanized 18 h after injection, and lymphocytes from tissues were harvested. The number of FITC⁺ cells in the spleen was used to normalize the amount of input and MLN, PP and lamina propria FITC⁺ cells were normalized to splenic FITC to compare DC gut homing to different organs in different animals as described[66].

## LPDC and IgD⁺ B cell isolation and co-culture

C57BL/6J mice were subcutaneously injected with B16 cells secreting Flt3-L as described[67]. After 12 to 14 days, mice were euthanized and the

small intestine lamina propria was digested as described above. Single-cell suspensions were incubated with mAbs to CD11c and IgA. Sorted LPDC (>95% CD11c$^+$ IgA$^-$) were resuspended at $2 \times 10^6$/ml and used immediately in co-culture assays. Naïve B cells were purified from the spleen by sorting IgD$^+$ IgA$^-$ cells (>95% IgD$^+$ IgA$^-$ cells). Sorted LPDC were treated with 100 uM serotonin (14927, Sigma) +/− 1uM HTR7 antagonist SB 269970 hydrochloride for 4 h and then lysed in Trizol for RNA extraction.

For co-cultures of sorted IgD$^+$ and LPDC cells, IgD$^+$ B cells were activated with 10 µg/ml anti-mouse IgM F(ab')2 (Poly21571, Biolegend), either alone or plus LPDC (1:1 ratio), treated with or without 100 uM serotonin (14927, Sigma) and/or 1uM HTR7 antagonist SB 269970 hydrochloride (1612, Tocris) for 7 days. DMSO was used as a mock treatment. Then the supernatant was collected for IgA measurement as above.

### In vivo HTR7 antagonist treatment
Co-housed C57BL/6J WT mice were treated with either intraperitoneal (i.p.) PBS or an HTR7 antagonist (SB269970, 10 mg/kg) every other day for 10 days. Fecal IgA measurements were carried out after the 10-day HTR7 antagonist treatment. After this treatment period, the mice were infected with *Salmonella*.

### In vivo STm infection
*Salmonella* Typhimurium (SL1344, STm) were grown for ~16 h at 37 °C with shaking and then subcultured (1:33) in lysogeny broth (LB) without antibiotics for 3 h until the cultures reached an optical density at 600 nm of 0.8. To prepare the inoculum, cultures were first pelleted at $5000 \times g$ for 5 min. The pellets were resuspended in PBS. For oral infection, mice were fasting for 4 h before being infected orogastrically with $5 \times 10^8$ STm suspended in 100 µl PBS. For intravenous infection, mice were infected with $1 \times 10^2$ STm suspended in 100 µl PBS via the lateral tail vein. Survival was monitored by daily observation and Kaplan–Meier survival graphs were generated by Prism software (GraphPad, version 9.4.1). STm colony-forming units (CFU) in the spleen and liver were measured 5 days after infection by plating serial dilutions of homogenized tissue samples on LB plates containing 100 µg/ml streptomycin.

### Cell-cell interaction analysis
We used the single-cell RNA-seq data generated from this study to analyze enteric neuron and LPDC interactions using the CellChat package[47]. Data analysis was performed with Seurat v4.3.0 as above except when the data was combined the resolution was set to 0.1 for FindClusters. CellChat identified differentially expressed ligands and receptors from each cell group and associated each interaction with a communication probability. Significant interactions were identified using default settings in CellChat. The results were visualized using bubble and chordal plots through the netVisual_bubble and chord-Diagram functions in CellChat. The CommunicationPatterns function in the CellChat package was used to identify and visualize receptor repertoires.

### Statistical methods
Statistical analyses were carried out using the two-tailed Student's *t* test, two-tailed Mann–Whitney test, or Kaplan–Meier Log-rank test on GraphPad Prism5 (version 9.4.1).

### Reporting summary
Further information on research design is available in the Nature Portfolio Reporting Summary linked to this article.

## Data availability
All scRNA-seq, bulk RNA-seq datasets and 16S sequencing data generated here have been deposited into the NCBI Gene Expression Omnibus database under accession number GSE227340. https://www.ncbi.nlm.nih.gov/geo/query/acc.cgi?acc=%20GSE227340 Source data are provided with this paper.

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

## Acknowledgements

We thank the members of the Waldor lab and Drs. Brandon Sit, Meenakshi Rao, Subhash Kulkarni, and Susan M. Dymecki for helpful discussions on all aspects of this project, Dr. Gerard Karsenty at Columbia University for *Tph2$^{-/-}$* mice, Dr. Ulrich H. Von Andrian at Harvard Medical School for the B16-FLT3L cell line. Schematics and model figures in Fig. 3H and supplementary Fig. 8D were generated with BioRender.com. Research in the M.K.W. laboratory is supported by HHMI and NIH grant R01 AI-042347.

## Author contributions

Conceptualization: H.L.Z., R.P.J. and M.K.W. Methodology: H.L.Z., Y.H., M.S., T.Z., and D.R.L. Investigation: H.L.Z., Y.H., M.S., T.Z., and D.R.L. Visualization: H.L.Z., Y.H., M.S., T.Z., and D.R.L. Funding acquisition: M.K.W. Supervision: M.K.W. Writing – original draft: H.L.Z. Writing – review & editing: H.L.Z. and M.K.W.

## Competing interests

The authors declare no competing interests.
