## [Peer Review File · Nature Communications]

Mouse enteric neurons control intestinal plasmacytoid dendritic cell function via serotonin-HTR7 signalingREVIEWER COMMENTS

Reviewer #1 (Remarks to the Author):

The manuscript by Zhang et al. describes their study on the role of neuronal serotonin in regulation of immune function, particularly dendritic cells (DCs) and IgA induction, and in host defense in *Salmonella* infection. The area of research is important, and the endeavour to link the neuronal serotonin and DCs cells in relation to host defense in intestinal bacterial infection is interesting. However, there are several issues which need to be addressed to enhance the quality of the manuscript.

Major Concerns:

1. Studies including a recent one from the authors of this manuscript demonstrated that reduction in neuronal serotonin reduces gut motility (Li Z et al. *J Neurosci.* 2011 Jun 15;31(24):8998-9009; Zang et al. *Curr Biol.* 2024 Feb 26;34(4):R133-R134). Reduced gut motility by decreased neuronal serotonin may increase pathogens burden and reduce survival. It will be important to investigate whether reduction of neuronal serotonin in Tph2 KO and conditional Tph2 KO mice has any effect on gut motility in the context of host defense in *Salmonella* infection.

2. Enterochromaffin (EC) cells are the main source of serotonin in the intestine and its production is regulated by Tph1 enzyme. The serotonin produced by enteric neurons (regulated by Tph2) is little as compared to the amount produced by EC cells. It is intriguing that the authors have observed differences in DCs function and IgA production in Tph2 conditional KO mice. Is there any difference in total intestinal serotonin levels in Tph2 KO and conditional KO mice compared to wild-type mice with or without *Salmonella* infection? Is there any role of EC cell derived serotonin in modulation of DC cell function and IgA?

3. The authors should include studies with 5-HT7 receptor KO mice to further define the role of this receptor in mediating the effect of neuronal serotonin. It will be also interesting to see whether irradiated mice reconstituted with bone marrow cells lacking 5-HT7 receptor expression have altered DC cell function and IgA production.

4. For all experiments involving *Salmonella* – were any further markers of infection severity measures (i.e. cytokine measures etc.) evaluated in intestinal tissue or systemically?

5. Microbial differences are very apparent between the sexes with WT and Tph2 KO mice. Was there any difference in *Salmonella* susceptibility in these studies between male and female mice?

6. Across experiments, there seems to be highly variable groups sizes across experiments – How were the studies powered?

7. Discussion seems incomplete and could be expanded to add more depth to the paper – in terms of how findings fit into the current literature and bigger implications of the work.

Minor Concerns:

1. May be prudent to include the full name of "DC" in the title.

2. The authors could include information on other published studies on 5-HT7 receptor on DCs in the intestine.

3. Line 24: please include full name of *Salmonella* strain here.

4. Line 38: Type of infection should be included here.

5. Line 42: Please include intestinal context here.

- 6.Line 44: Awkward wording, something may be missing here.
- 7.Line 48-50: Awkwardly worded, perhaps specifying the CNS would be prudent here.
- 8.Line 55-57: Ending of this paragraph seems out of place with the rest of the paragraph. Perhaps a transition would be prudent or moving this to another place in the introduction.
- 9.Line 61: Needs context, "in the gut."
- 10.Line 64-66: Unclear/awkward wording, rework suggested.
- 11.Line 68-69: Line contradicts ending of the paragraph.
- 12.Line73: Please include full name of ILC.
- 13.Line 77-80: This line seems like a very tenuous connection to report without more context. Perhaps remove or reword for clarity/increased justification.
- 14.Line 83: Please specify which circuits are being referred too.
15. Use of the word "controls" (Line 97, 100 etc.) in several headings/ throughout the paper seems like strong language to use in this context/ based on the data presented.
- 16.Line 101-108: Experiments involving SERT mice may be more appropriate in supplemental and seem a bit extraneous here.
- 17.Line 118: Salmonella italics missing.
- 18.Line 338: It would be interesting to tie the findings full circle – Were any "immune ligands" as described in Line 305- 313, measured in Salmonella infection experiments?
- 19.Line 574: "critical" again seems like a strong word here.
- 20.Line 636: n numbers are different between the different analyses in the figure 3D please correct in text.
- 21.Line 734-737: Please include Author et al. to end these sentences – unclear what is being referred too.
- 22.Line 862,920: Unneeded capitalization
- 23.Throughout the paper and figure descriptions, formatting of HTR7 is different and should be kept consistent.

Comments Specific for the Figures:

- 1.For figures showing FACS please provide clearer images (Figure S1D etc.)
- 2.No scale bar is apparent in Figure 2E.
- 3.Supplementary figure 4H appears rather random within the context of figure S4. It may be more appropriate to make this a separate supplementary figure.

Reviewer #2 (Remarks to the Author):

The manuscript submitted focuses on the neural-immune axis, addressing the importance of

serotonin (5-HT)-producing enteric neurons regulating innate/adaptive responses and control of infection. The authors engineered conditional mice with *tph2*^{-/-} enteric neurons and identified a HTR7-dependent interaction mechanism with and regulation of intestinal plasmacytoid dendritic cells (pDCs) that control IgA levels and susceptibility to Salmonella infections. The topic of the manuscript is significant and timely due to the increasing evidence that suggests a biologically relevant bidirectional connection between the neural and immune systems in the intestinal environment. The manuscript is clearly written, and the results shown are compelling. The findings are novel. The protocols used, as well as the statistical analysis used, are appropriate and well described in detail. Sequencing data (scRNA-seq, bulk RNA-seq, and 16S seq are available).

There is a reduction of fecal IgA+ in *Thp2*^{-/-} mice. The authors state that knockout and WT mice microbiota were similar (Fig S1F). However, only a visual characterization of relative abundances obtained from a small number of mice (n=6) is shown, and no statistical parameters are provided. It is uncertain whether the microbiota is similar or not. This is an important factor due to the role of IgA in controlling the commensal microbiota composition and how relevant the microbiota is in regulating susceptibility to infection. More specifically to the manuscript's topic, 5-HT is produced by specific microbiota taxa. It is unknown whether the knockout strains (whole and conditional knockout) have an altered 5-HT-producing microbiota.

In Figure 1, the accumulation of serotonin in *Sert*^{-/-} is not shown.

The homing assay used is designed from DCs, which may not be able to be translated to pDCs.

Reviewer #3 (Remarks to the Author):

In the manuscript by Zhang et al investigates the role of serotonin signaling in gut mucosal immune function. Using different mouse models, they demonstrate that neuronal serotonin promotes IgA class switching and immune defense against oral Salmonella infection. They further demonstrate that this response is mediated by neuronal crosstalk with plasmacytoid dendritic cells. The study is comprehensive in its approach with mouse models, genetic manipulation, gene expression analysis, bioinformatics and spatial validation. The conclusions are well supported, however the clinical significance of these findings is not well established.

Major:

The authors have not attempted to relate their findings to human biology. At a minimum, are equivalent expressions of HTR7 and ligand-receptors of mouse pDCs seen in published single-cell RNA sequencing datasets of human colon?

The potential biological significance of enteric serotonergic neurons regulating intestinal immune cell function and vice versa is missing from the discussion.

To further support the role of pDC-ESN communication in immune defence against pathogens, the IgA response and pDC phenotype could have been assessed in the *Sert*^{-/-} mouse, which the authors demonstrate has increased survival.

Minor points:

Figure 3G: a vehicle control should be included in this in vitro experiment.

Figure 3E,F: The results would be easier to interpret if an arrow or label was used to show what the fold change is in relation to. For example, an arrow above the volcano plot pointing right with 'serotonin-induced'.

RESPONSE TO REVIEWERS' COMMENTS

REVIEWER 1

The manuscript by Zhang et al. describes their study on the role of neuronal serotonin in regulation of immune function, particularly dendritic cells (DCs) and IgA induction, and in host defense in *Salmonella* infection. The area of research is important, and the endeavour to link the neuronal serotonin and DCs cells in relation to host defense in intestinal bacterial infection is interesting. However, there are several issues which need to be addressed to enhance the quality of the manuscript.

Thank you for critique and interest in our work.

Major Concerns:

1. Studies including a recent one from the authors of this manuscript demonstrated that reduction in neuronal serotonin reduces gut motility (Li Z et al. *J Neurosci.* 2011 Jun 15;31(24):8998-9009; Zang et al. *Curr Biol.* 2024 Feb 26;34(4):R133-R134). Reduced gut motility by decreased neuronal serotonin may increase pathogens burden and reduce survival. It will be important to investigate whether reduction of neuronal serotonin in *Tph2* KO and conditional *Tph2* KO mice has any effect on gut motility in the context of host defense in *Salmonella* infection.

Thanks for raising this interesting point. As noted, there was reduced gut motility in our conditional *Tph2* KO animal (PMID: 38412819) and we agree that reduced gut motility could contribute to the increased *Salmonella* burden. As suggested, we measured gut motility before and after *Salmonella* infection in both WT and *Tph2* KO mice. Consistent with our findings published in *Current Biology* (PMID: 38412819), *Tph2* KO mice exhibited decreased gut motility compared to WT littermate controls. Interestingly, *Salmonella* infection significantly increased intestinal motility in WT mice, although the underlying mechanism remains unclear. *Tph2* KO animals also exhibited a similar increase in gut motility with *Salmonella* infection (Fig R1 below and Fig S6A in the paper). Drawing definitive conclusions about the impact of gut motility on *Salmonella* infection phenotypes based on this data is challenging.

To further investigate whether changes in gut motility account for increased *Salmonella* burdens in distal organs, we measured the effect of an HTR7 antagonist on gut motility, *Salmonella* dissemination to distal organs and IgA B cell differentiation. We found that animals treated with the HTR7 antagonist phenocopied the conditional *Tph2* KO mice in that they exhibited decreased levels of fecal IgA (Fig R1C, now included in the revised manuscript as Fig 3J) and increased susceptibility to *Salmonella* infection (Fig R1D, now included in the revised manuscript as Fig 3K). However, these HTR7 antagonist-treated mice did not exhibit diminished gut motility (Fig R1B; now included in the revised manuscript Fig S6B). These findings provide strong support for the proposed enteric serotonergic neuron-pDC-IgA-*Salmonella* susceptibility circuitry and argue that the gut motility defect in the *Tph2* KO animals do not explain their immune deficits and susceptibility to *Salmonella*. These data not only further support our hypothesis but also open a novel avenue for drug discovery in immune regulation.

Fig R1. Mice treated with an HTR7 antagonist exhibit normal gut motility but reduced fecal IgA levels and increased dissemination of *Salmonella* to distal organs.

(A) Total GI transit time was measured using a carmine red assay in WT and *Tph2*^{-/-} before and after *Salmonella* infection.

(B) Gut transit time in mock (PBS) and HTR7 antagonist-treated mice. The mice were treated with ip PBS or an HTR7 antagonist (SB269970) 15 minutes prior to the measurement of gut transit time with carmine red assay.

(C) Fecal IgA levels in mock and HTR7 antagonist-treated mice. The mice were treated with either ip PBS or an HTR7 antagonist every other day for 10 days. Fecal IgA measurements were carried out after the 10-day HTR7 antagonist treatment.

(D) *Salmonella* CFU burdens in spleens and livers of mock and HTR7 antagonist-treated mice 5 days post oral inoculation. The mice were treated as described in (C) before oral *Salmonella* infection.

Data shown are means ± SD. Statistical analysis was performed by a two-tailed Mann-Whitney test.

2. Enterochromaffin (EC) cells are the main source of serotonin in the intestine and its production is regulated by Tph1 enzyme. The serotonin produced by enteric neurons (regulated by Tph2) is little as compared to the amount produced by EC cells. It is intriguing that the authors have observed differences in DCs function and IgA production in *Tph2* conditional KO mice. Is there any difference in total intestinal serotonin levels in *Tph2* KO and conditional KO mice compared to wild-type mice with or without *Salmonella* infection? Is there any role of EC cell derived serotonin in modulation of DC cell function and IgA?

These are all interesting questions. Brommage et al. showed that intestinal content serotonin levels are very similar in WT and whole-body *Tph2* KO animals (PMID: 26229596). Interestingly, despite this similarity, *Tph2*-defective animals exhibit heightened inflammation in the context of DSS colitis (23303993), whereas *Tph1*-defective animals display reduced inflammation (19706294). These findings suggest that the source (enterochromaffin vs neuron, and presumably the spatial distribution) of serotonin in the gut has a critical impact on its physiological roles. Enteric neuron-derived serotonin may influence DC function in a highly spatially restricted synapse-like manner; in contrast the effects of EC-derived serotonin are likely not as spatially restricted.

We fully agree that the role of EC cell derived serotonin in modulation of DC and B cell function warrants further investigation and we are in the process of generating intestine epithelial cell-specific *Tph1* conditional KO mice to delve deeper into this matter, but in our view these studies are beyond the scope of this manuscript.

3. The authors should include studies with 5-HT7 receptor KO mice to further define the role of this

receptor in mediating the effect of neuronal serotonin. It will be also interesting to see whether irradiated mice reconstituted with bone marrow cells lacking 5-HT7 receptor expression have altered DC cell function and IgA production.

We agree this is an excellent experiment; however, HTR7 KO mice only exist as frozen sperm and creating the mice would require at least 6-9 months; furthermore, floxed HTR7 animals are not available and creation of mice deficient in HTR7 in DCs is not feasible at the point. As discussed in point 1, to further establish the role of HTR7 in mediating the effects of serotonin, we used the HTR7 antagonist SB269970. We found that animals treated with the HTR7 antagonist showed decreased levels of fecal IgA and increased susceptibility to *Salmonella* infection. These findings provide strong support for the proposed enteric serotonergic neuron-pDC-IgA-*Salmonella* susceptibility circuitry.

4. For all experiments involving *Salmonella* – were any further markers of infection severity measures (i.e. cytokine measures etc.) evaluated in intestinal tissue or systemically?

Thank you for your comment. Since CFU and survival are well-documented indicators of infection severity, cytokines in the blood and intestine were not measured. In addition, we would like to emphasize that our focus was on the role of serotonin produced by enteric neurons in modulating immune development and function and not the specific mechanisms by which deficiencies in *Tph2* result in susceptibility to *Salmonella*.

5. Microbial differences are very apparent between the sexes with WT and *Tph2* KO mice. Was there any difference in *Salmonella* susceptibility in these studies between male and female mice?

We indeed observed differences in microbiota composition between male and female mice (Fig S1 I,K). Additionally, previous studies have noted sex-specific responses to various pathogens, such as *Listeria* and *Streptococcus* (PMID: 16113316, 23166492). However, we did not observe any differences between male and female mice in their response to oral *Salmonella* infection, which aligns with previous findings (PMID: 35417454, Fig 2).

6. Across experiments, there seems to be highly variable groups sizes across experiments – How were the studies powered?

Thanks for raising this point. In all of our experiments, we relied on co-housed litter mate controls that were offspring of heterozygous parents. This approach, which assiduously controls for litter and microbiota effects, can result in variability in animal numbers. We repeated all of our studies with at least 3 litters and the statistical analyses are presented in all cases.

Our power calculations indicate that we had sufficient numbers of animals in our experiments. For instance, in our CFU analyses, we expect a 100-fold difference between means and a 1 log standard deviation, indicating that we would need 9 animals to achieve a $p < 0.01$ with 95% probability. In our immune cell analysis, we expect a 3-fold difference between means and 40% standard deviation, indicating that we would need 7 animals to achieve a $p < 0.01$ with 95% probability. Thus, we used more than the minimal number of animals for different assays.

7. Discussion seems incomplete and could be expanded to add more depth to the paper – in terms of how findings fit into the current literature and bigger implications of the work.

We have expanding the scope of our discussion. In particular, we now include new references and discussion about human scRNA-seq data revealing that intestinal DCs express HTR7. These data provide support for the idea that our findings likely have significance for understanding the roles of serotonin-HTR7 signalling in human immune regulation (lines 396-400).

Minor Concerns:

Thanks for these excellent suggestions; we modified the manuscript accordingly.

1. May be prudent to include the full name of “DC” in the title.

done

2. The authors could include information on other published studies on 5-HT7 receptor on DCs in the intestine.

Thanks, we have incorporated the changes into the discussion.

3. Line 24: please include full name of Salmonella strain here.

Changed.

4. Line 38: Type of infection should be included here.

Changed.

5. Line 42: Please include intestinal context here.

Changed.

6. Line 44: Awkward wording, something may be missing here.

We eliminated this line and the other 3 'highlights'.

7. Line 48-50: Awkwardly worded, perhaps specifying the CNS would be prudent here.

Changed.

8. Line 55-57: Ending of this paragraph seems out of place with the rest of the paragraph. Perhaps a transition would be prudent or moving this to another place in the introduction.

Thanks, we agree. We have removed this sentence from the introduction and use it now in the Discussion to expand our discussion of the potential utility of specific serotonin receptor antagonists in modulation of immune function.

9. Line 61: Needs context, “in the gut.”

Changed.

10. Line 64-66: Unclear/awkward wording, rework suggested

Re-worded.

11. Line 68-69: Line contradicts ending of the paragraph.

Re-worded.

12. Line 73: Please include full name of ILC.

Thanks, we've added the full name of innate lymphoid cells (ILCs) here.

13. Line 77-80: This line seems like a very tenuous connection to report without more context.

Perhaps remove or reword for clarity/increased justification.

Re-worded for clarity.

14. Line 83: Please specify which circuits are being referred to.

Changed as suggested.

15. Use of the word “controls” (Line 97, 100 etc.) in several headings/ throughout the paper seems like strong language to use in this context/ based on the data presented.

Thank you, changes implemented per your feedback.

16. Line 101-108: Experiments involving SERT mice may be more appropriate in supplemental and seem a bit extraneous here.

Thanks, we have moved this part to the supplemental data.

17. Line 118: Salmonella italics missing.

Revised, and thank you for your thorough review.

18.Line 338: It would be interesting to tie the findings full circle – Were any “immune ligands” as described in Line 305- 313, measured in Salmonella infection experiments?

Thanks, we are currently investigating the role of enteric neuron-derived I17 in host defense.

19.Line 574: “critical” again seems like a strong word here.

Thank you, changes implemented per your feedback.

20.Line 636: n numbers are different between the different analyses in the figure 3D please correct in text.

Thanks, corrected.

21.Line 734-737: Please include Author et al. to end these sentences – unclear what is being referred to.

Thanks, revised as suggested.

22.Line 862,920: Unneeded capitalization

Thanks, revised as suggested

23.Throughout the paper and figure descriptions, formatting of HTR7 is different and should be kept consistent.

Thanks, revised as suggested.

Comments Specific for the Figures:

1.For figures showing FACS please provide clearer images (Figure S1D etc.)

Thank you for the feedback. The original data has been uploaded for review.

2.No scale bar is apparent in Figure 2E.

Thanks, the scale bar was added.

3.Supplementary figure 4H appears rather random within the context of figure S4. It may be more appropriate to make this a separate supplementary figure.

Thanks, revised as suggested.

REVIEWER 2

The manuscript submitted focuses on the neural-immune axis, addressing the importance of serotonin (5-HT)-producing enteric neurons regulating innate/adaptive responses and control of infection. The authors engineered conditional mice with *tph2*^{-/-} enteric neurons and identified a HTR7-dependent interaction mechanism with and regulation of intestinal plasmacytoid dendritic cells (pDCs) that control IgA levels and susceptibility to Salmonella infections. The topic of the manuscript is significant and timely due to the increasing evidence that suggests a biologically relevant bidirectional connection between the neural and immune systems in the intestinal environment. The manuscript is clearly written, and the results shown are compelling. The findings are novel. The protocols used, as well as the statistical analysis used, are appropriate and well described in detail. Sequencing data (scRNA-seq, bulk RNA-seq, and 16S seq are available).

Thanks for the positive feedback; we really appreciate it.

There is a reduction of fecal IgA⁺ in *Thp2*^{-/-} mice. The authors state that knockout and WT mice microbiota were similar (Fig S1F). However, only a visual characterization of relative abundances obtained from a small number of mice (n=6) is shown, and no statistical parameters are provided. It is uncertain whether the microbiota is similar or not. This is an important factor due to the role of IgA in controlling the commensal microbiota composition and how relevant the microbiota is in regulating susceptibility to infection. More specifically to the manuscript's topic, 5-HT is produced by specific microbiota taxa. It is unknown whether the knockout strains (whole and conditional knockout) have an altered 5-HT-producing microbiota.

Thanks for raising these valid points. We conducted statistical analyses of the microbiota data (presented in Fig S1 I and K) and found that the differences between WT and KO were very modest (Fig. R2 and Fig S1 J and L in the revised manuscript). We note that our experiments were always carried with co-housed littermate controls that were the offspring of heterozygous breeders; since mice are coprophagic we did not anticipate major differences in the microbiota using this experimental scheme. As mentioned above (Reviewer 1, point 5), there are significant differences in microbiota composition between male and female mice, however our data, along with previous findings (PMID: 35417454, Fig 2) suggest that these sex-linked microbiota differences do not alter susceptibility to oral *Salmonella* infection. We suspect that the mild difference in microbiota between WT and KO animals is not the determining factor for *Salmonella* susceptibility.

We suspect that the minimal changes in the microbiota in WT and KO animals do not significantly contribute to serotonin production because the taxa that differ are not major contributors to serotonin production according to the most recent study (PMID: 38489352).

Fig R2:

Fig R2. Differential abundance of fecal microbiota at the OTU level between WT and *Tph2* KO animals, as well as between Hand2 Cre-positive and negative animals. Differential abundance of OTUs was analyzed using R package ANCOMBC (PMID: 32665548).

In Figure 1, the accumulation of serotonin in *Sert*^{-/-} is not shown.

Thanks for this comment. Several studies have demonstrated the accumulation of extracellular serotonin in *Sert* KO animals (PMID: 32521224, Fig 3A; 30498565, Fig 3B; 25860609, Fig S4A). Additionally, since *Sert* KO is not the main focus of this study, we have moved this aspect to the supplemental data as suggested by Reviewer 1.

The homing assay used is designed from DCs, which may not be able to be translated to pDCs.

This is an interesting point. However, our scRNA-seq show the pDC and not other DCs express CCR7. Since we show that enteric *Tph2* positive neurons mediated DC homing is dependent on both HTR7 and CCR7 (Fig 3I), our findings suggest that we are measuring pDC gut homing.

Reviewer #3 (Remarks to the Author):

In the manuscript by Zhang et al investigates the role of serotonin signaling in gut mucosal immune function. Using different mouse models, they demonstrate that neuronal serotonin promotes IgA class switching and immune defense against oral *Salmonella* infection. They further demonstrate that this response is mediated by neuronal crosstalk with plasmacytoid dendritic cells. The study is comprehensive in its approach with mouse models, genetic manipulation, gene expression analysis, bioinformatics and spatial validation. The conclusions are well supported, however the clinical significance of these findings is not well established.

Thank you for the positive comments.

Major:

The authors have not attempted to relate their findings to human biology. At a minimum, are equivalent expressions of HTR7 and ligand-receptors of mouse pDCs seen in published single-cell RNA sequencing datasets of human colon?

Thanks for this raising this intriguing issue. To begin to address this question, we delved into published human single-cell RNA sequencing data from studies on human colorectal cancer (PMID: 34450029), Crohn's disease (PMID: 33771991), and pediatric functional gastrointestinal disorders (eLife.91792.1). We found that, similar to mice, HTR7 is the exclusive serotonin receptor expressed by human DCs (Fig R4). We now discuss the similarity of human and mice serotonin receptor expression in our revised discussion (lines 396-400 and Fig S6C). Although the proportion of HTR7+ DC appears low in the human datasets, we think that low sequencing depth could account for these numbers and further investigation of the expression and roles of HTR7 in human biology is warranted.

Fig R4. Expression of different serotonin receptors in the human colon cancer atlas. The data was visualized using Single Cell Portal with the parameters: clustering (all cells), annotation (ClusterMidWay), subsampling (all cells), and gene collapse (none). This analysis reveals that proportion of human dendritic cells and macrophages express *Htr7* mRNA.

The potential biological significance of enteric serotonergic neurons regulating intestinal immune cell function and vice versa is missing from the discussion.

Thank you for your comment. We have added more discussion about the potential biological significance of this novel neuron-immune interaction, however, we do not want to be overly speculative (lines 397-400).

To further support the role of pDC-ESN communication in immune defence against pathogens, the IgA response and pDC phenotype could have been assessed in the Sert^{-/-} mouse, which the authors demonstrate has increased survival.

Thanks for raising this point. We carried out more experiment and found that animals treated with the HTR7 antagonist showed decreased levels of fecal IgA and increased susceptibility to *Salmonella* infection (Reviewer 1, pt 1). These results further support the role of ESN-pDC communication in immune defence against pathogens.

Sert has been studied for its role in host defense against pathogens (PMID: 32521224). However, despite the apparent consistency in infection phenotypes between *Tph2* KO (reduced serotonin and increased susceptibility) and *Sert* KO (increased serotonin and decreased susceptibility) animals, elucidation of the mechanisms of host defense in *Sert* KO mice is challenging because *Sert* is broadly expressed across various cell types, including neuronal and non-neuronal cells. Therefore, we have chosen to concentrate on neuron-derived serotonin using conditional *Tph2* KO animals. Nevertheless, we believe that *Sert* KO warrants further investigation, but it is beyond the scope of this study.

Minor points:

Figure 3G: a vehicle control should be included in this in vitro experiment.

Thank you for this comment. We used a DMSO control in this experiment, and it showed no effect on IgA induction. We note that in this experiment very little DMSO (1:10,000 dilution) is present. We now mention this control in the methods.

Figure 3E, F: The results would be easier to interpret if an arrow or label was used to show what the fold change is in relation to. For example, an arrow above the volcano plot pointing right with 'serotonin-induced'.

Thank you and changed accordingly.

REVIEWER COMMENTS

Reviewer #1 (Remarks to the Author):

The authors have reasonably addressed most of my concerns. Nevertheless, inclusion of new data on motility and 5-HT7 antagonists raised additional issues related to interpretation of findings and conclusion which need to be addressed to help improve and provide clarity to the manuscript:

1) In the revised manuscript the authors included data to show motility in Tph2 KO and wild-type mice with or without Salmonella infection. Tph2 KO mice exhibited reduced gut motility compared to WT controls without infection. Salmonella infection significantly increased intestinal motility in WT mice and the authors mentioned that Tph2 KO animals also exhibited a similar increase in gut motility after infection. Although there is an increase in motility in Tph2 KO the figure clearly shows that there is significant difference in motility between the Tph2 KO and WT mice after infection which may influence susceptibility to infection.

2) 5-HT7 antagonist was used to phenocopy the conditional Tph2 KO mice. However, treatment with 5-HT7 antagonist will block the effect of both mucosal and neuronal serotonin. It is not clear how the authors concluded that they have blocked neuronal serotonin only.

3) The authors should include discussion on future work on the role of EC cell derived/mucosal serotonin in modulation of DC and B cell function.

Reviewer #2 (Remarks to the Author):

The authors addressed my comments appropriately. I don't have further suggestions.

Reviewer #3 (Remarks to the Author):

The authors have appropriately responded to my initial comments and the manuscript has been improved through the review process. I have no further comments.

REVIEWER COMMENTS

Reviewer #1 (Remarks to the Author):

The authors have reasonably addressed most of my concerns. Nevertheless, inclusion of new data on motility and 5-HT7 antagonists raised additional issues related to interpretation of findings and conclusion which need to be addressed to help improve and provide clarity to the manuscript:

1) In the revised manuscript the authors included data to show motility in Tph2 KO and wild-type mice with or without *Salmonella* infection. Tph2 KO mice exhibited reduced gut motility compared to WT controls without infection. *Salmonella* infection significantly increased intestinal motility in WT mice and the authors mentioned that Tph2 KO animals also exhibited a similar increase in gut motility after infection. Although there is an increase in motility in Tph2 KO the figure clearly shows that there is significant difference in motility between the Tph2 KO and WT mice after infection which may influence susceptibility to infection.

Thanks for the comment; we agree that gut motility could be one reason for *Salmonella* susceptibility in *Tph2* KO mice, thus we further clarified this point in the discussion (lines 357-360) stating that:

“However, even in the setting of infection, the *Tph2* KO animals had reduced gut motility compared to the WT mice. Thus, the gut motility defect observed in *Tph2* KO animals could contribute to their susceptibility to *Salmonella*.”

2) 5-HT7 antagonist was used to phenocopy the conditional Tph2 KO mice. However, treatment with 5-HT7 antagonist will block the effect of both mucosal and neuronal serotonin. It is not clear how the authors concluded that they have blocked neuronal serotonin only.

We do not claim that HTR7 antagonist specifically blocks the action of serotonin produced by neurons. To clarify this point, we revised this paragraph (lines 253-256) to read:

“Although the HTR7 antagonist could also block the effects of serotonin produced by cells other than neurons, these findings provide additional support for the role of the HTR7 signaling pathway in the pDC-IgA-*Salmonella* susceptibility circuitry.”

3) The authors should include discussion on future work on the role of EC cell derived/mucosal serotonin in modulation of DC and B cell function.

Thanks for raising this interesting point. We are currently investigating this aspect using *Vil-Cre*; *Tph1^{fl/fl}* animals and have added the following sentence to the discussion lines (394-395):

“Furthermore, it will be intriguing to investigate whether serotonin derived from enterochromaffin cells affects DC and B cell function.”

Reviewer #2 (Remarks to the Author):

The authors addressed my comments appropriately. I don't have further suggestions.
Thank you!

Reviewer #3 (Remarks to the Author):

The authors have appropriately responded to my initial comments and the manuscript has been improved through the review process. I have no further comments.

We sincerely thank all the reviewers once again for their invaluable support and efforts, which have significantly enhanced the impact and clarity of our work, making it more compelling and robust for readers.